# ISP: Multi-Layered Garment Draping with Implicit Sewing Patterns

**Ren Li**     **Benoît Guillard**     **Pascal Fua**
Computer Vision Lab, EPFL
Lausanne, Switzerland
`ren.li@epfl.ch`   `benoit.guillard@epfl.ch`   `pascal.fua@epfl.ch`

## Abstract

Many approaches to draping individual garments on human body models are realistic, fast, and yield outputs that are differentiable with respect to the body shape on which they are draped. However, they are either unable to handle multi-layered clothing, which is prevalent in everyday dress, or restricted to bodies in T-pose. In this paper, we introduce a parametric garment representation model that addresses these limitations. As in models used by clothing designers, each garment consists of individual 2D panels. Their 2D shape is defined by a Signed Distance Function and 3D shape by a 2D to 3D mapping. The 2D parameterization enables easy detection of potential collisions and the 3D parameterization handles complex shapes effectively. We show that this combination is faster and yields higher quality reconstructions than purely implicit surface representations, and makes the recovery of layered garments from images possible thanks to its differentiability. Furthermore, it supports rapid editing of garment shapes and texture by modifying individual 2D panels.

## 1   Introduction

Draping virtual garments on body models has many applications in fashion design, movie-making, virtual try-on, virtual and augmented reality, among others. Traditionally, garments are represented by 3D meshes and the draping relies on physics-based simulations (PBS) [1, 2, 3, 4, 5, 6, 7, 8, 9, 10, 11, 12] to produce realistic interactions between clothes and body. Unfortunately PBS is often computationally expensive and rarely differentiable, which limits the scope of downstream applications. Hence, many recent techniques use neural networks to speed up the draping and to make it differentiable. The garments can be represented by 3D mesh templates [13, 14, 15, 16, 17, 18, 19, 20], point clouds [21, 22], UV maps [23, 24, 25, 26, 27], or implicit surfaces [28, 29, 30]. Draping can then be achieved by Linear Blend Skinning (LBS) from the shape and pose parameters of a body model, such as SMPL [31].

Even though all these methods can realistically drape individual garments over human bodies, none can handle multiple clothing layers, despite being prevalent in everyday dress. To address overlapping clothing layers such as those in Fig. 1 while preserving expressivity, we introduce an Implicit Sewing Pattern (ISP), a new representation inspired by the way fashion designers represent clothes. As shown in Fig. 2, a sewing pattern is made of several 2D panels implicitly represented by signed distance functions (SDFs) that model their 2D extent and are conditioned on a latent vector $\mathbf{z}$, along with information about how to stitch them into a complete garment. To each panel is associated a 2D to 3D mapping representing its 3D shape, also conditioned on $\mathbf{z}$. The 2D panels make it easy to detect collisions between surfaces and to prevent interpenetrations. In other words, the garment is made of panels whose 2D extent is learned instead of having to be carefully designed by a human and whose

37th Conference on Neural Information Processing Systems (NeurIPS 2023).

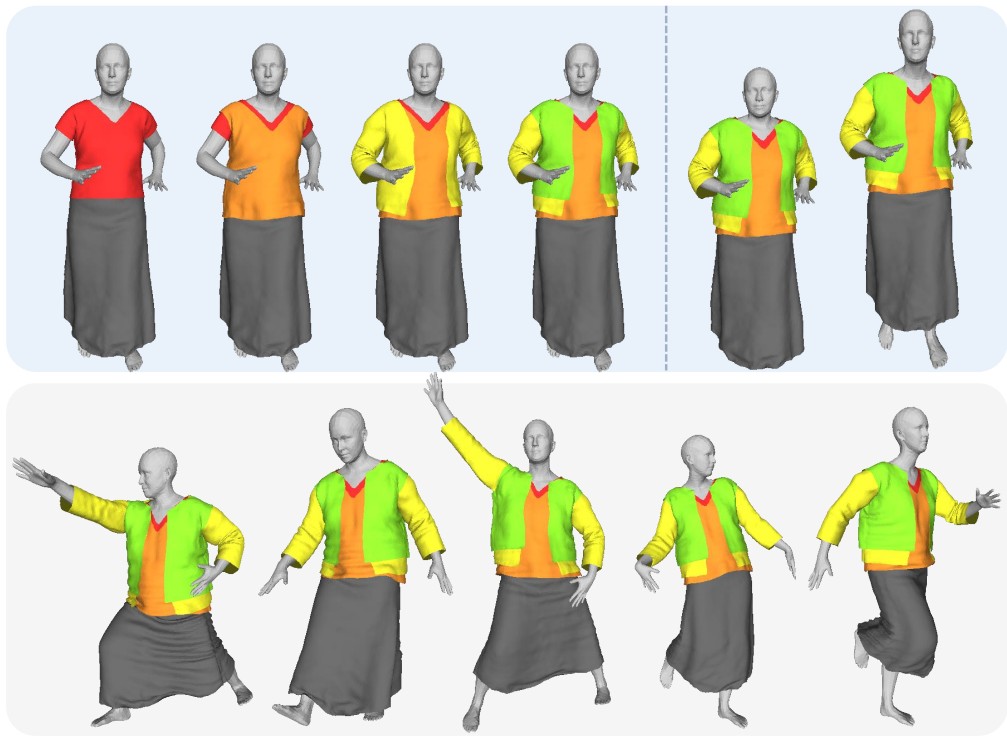

Figure 1: **Multi-layered garment draping**. **Top**: Draping multiple layers of garments over one body (left) and modifying the body's shape (right). **Bottom**: Draping the same set of 5 garments over bodies with varying poses and shapes.

3D shape is expressed in a unified $uv$ space in which a loss designed to prevent interpenetrations can easily be written.

This combination enables us to model layered garments such as those of Fig. 1 while preserving end-to-end differentiability. This lets us drape them realistically over bodies in arbitrary poses, to recover them from images, and to edit them easily. Doing all this jointly is something that has not yet been demonstrated in the Computer Vision or Computer Graphics literature. Furthermore, most data driven draping methods rely on synthetic data generated with PBS for supervision purposes. In contrast, ISPs rely on physics-based self-supervision of [18, 13]. As a result, at inference time, our approach can handle arbitrary body poses, while only requiring garments draped over bodies in a canonical pose at training time. Our code is available at `https://github.com/liren2515/ISP`.

## 2   Related Work

**Garment Draping.**    Garment draping approaches can be classified as physics-based or data-driven. Physics-based ones [3, 32, 33, 34, 35, 12] produce high-quality results but are computationally demanding, while data-driven approaches are faster, sometimes at the cost of realism. Most data-driven methods are template-based [13, 14, 15, 17, 19, 18, 20, 16, 18, 13], with a triangulated mesh modeling a specific garment and a draping function trained specifically for it. As this is impractical for large garment collections, some recent works [22, 21, 36] use 3D point clouds to represent garments instead. Unfortunately, this either prevents differentiable changes in garment topology or loses the point connections with physical meanings. The approaches of [24, 25] replace the clouds by UV maps that encode the diverse geometry of garments and predict positional draping maps. These UV maps are registered to the body mesh, which restricts their interpretability and flexibility for garment representation and manipulation. The resulting garments follow the underlying body topology, and the ones that should not, such as skirts, must be post-processed to remove artifacts. Yet other algorithms [29, 30, 37] rely on learning 3D displacement fields or hierarchical graphs for garment deformation, which makes them applicable to generic garments.

While many of these data-driven methods deliver good results, they are typically designed to handle a single garment or a top and a bottom garment with only limited overlap. An exception is the approach of [38] that augments SDFs with covariant fields to untangle multi-layered garments. However, the untangling is limited to garments in T-pose. In contrast, our proposed method can perform multi-layered garment draping for bodies in complex poses and of diverse shapes.

**Garments as Sets of Panels.** Sewing patterns are widely used in garment design and manufacturing [39, 40, 41, 42, 43]. Typically, flat sewing patterns are assembled and then draped using PBS. To automate pattern design, recent works introduce a parametric pattern space. For example, the methods of [44, 45] introduce a sparse set of parameters, such as sleeve length or chest circumference, while [27] relies on principal component analysis (PCA) to encode the shape of individual panels. It requires hierarchical graphs on groups of panels to handle multiple garment styles. By contrast, our approach relies on the expressivity of 2D Sign Distance Functions (SDFs) to represent the panels.

To promote the use of sewing patterns in conjunction with deep learning, a fully automatic dataset generation tool is proposed in [46]. It randomly samples parameters to produce sewing patterns and uses PBS to drape them on a T-posed human body model to produce training pairs of sewing patterns and 3D garment meshes.

**Garments as Implicit Surfaces.** SDFs have become very popular to represent 3D surfaces. However, they are primarily intended for watertight surfaces. A standard way to use them to represent open surfaces such as garments is to represent them as thin volumes surrounding the actual surface [28, 29], which can be represented by SDFs but with an inherent accuracy loss. To address this issue, in [30], the SDFs are replaced by unsigned distance functions (UDFs) while relying on the differentiable approach of [47] to meshing, in case an actual mesh is required for downstream tasks. However, this requires extra computations for meshing and makes training more difficult because surfaces lie at a singularity of the implicit field. This can result in unwanted artifacts that our continuous UV parameterization over 2D panels eliminates.

## 3 Method

In clothing software used industrially [39, 40, 41], garments are made from sets of 2D panels cut from pieces of cloth which are then stitched together. Inspired by this real-world practice, we introduce the Implicit Sewing Patterns (ISP) model depicted by Fig. 2. It consists of 2D panels whose shape is defined by the zero crossings of a function that takes as input a 2D location $\mathbf{x} = (x_u, x_v)$ and a latent vector $\mathbf{z}$, which is specific to each garment. A second function that also takes $\mathbf{x}$ and $\mathbf{z}$ as arguments maps the flat 2D panel to the potentially complex 3D garment surface within the panel, while enforcing continuity across panels. Finally, we train draping networks to properly drape multiple garments on human bodies of arbitrary shapes and poses, while avoiding interpenetrations of successive clothing layers.

### 3.1 Modeling Individual Garments

We model garments as sets of 2D panels stitched together in a pre-specified way. Each panel is turned into a 3D surface using a $uv$ parameterization and the networks that implement this parameterization are trained to produce properly stitched 3D surfaces. To create the required training databases, we use the PBS approach of [46]. Because it also relies on 2D panels, using its output to train our networks has proved straightforward, with only minor modifications required.

**Flat 2D Panels.** We take a pattern $P$ to be a subset of $\Omega = [-1, 1]^2$ whose boundary are the zero crossings of an implicit function. More specifically, we define

$$\mathcal{I}_\Theta : \Omega \times \mathbb{R}^{|\mathbf{z}|} \to \mathbb{R} \times \mathbb{N}, \ (\mathbf{x}, \mathbf{z}) \mapsto (s, c) \,, \tag{1}$$

where $\mathcal{I}_\Theta$ is implemented by a fully connected network. It takes as input a local 2D location $\mathbf{x}$ and a global latent vector $\mathbf{z}$ and returns two values. The first, $s$, should approximate the signed distance to the panel boundary. The second, $c$, is a label associated to boundary points and is used when assembling the garment from several panels. Then, boundaries with the same labels should be stitched together. Note that the $c$ values are only relevant at the panel boundaries, that is, when $s = 0$. However, training the network to produce such values *only* there would be difficult because this would involve a very sparse supervision. So, instead, we train our network to predict the label

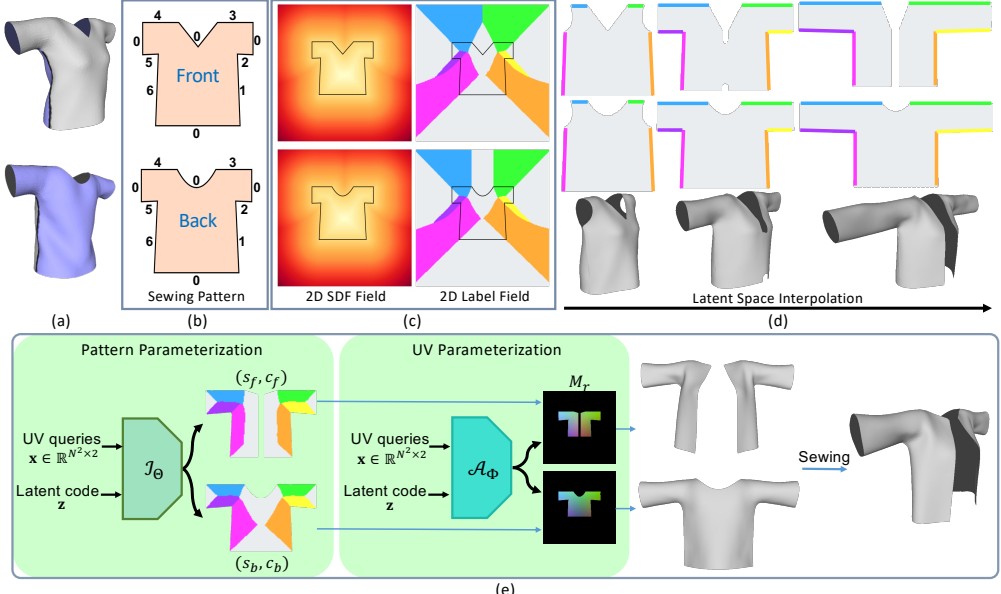

Figure 2: **Implicit Sewing Pattern (ISP). (a)** The 3D mesh surface for a shirt with the front surface in gray and the back one in blue. **(b)** The front and back 2D panels of the ISP. The numbers denote the labels of seamed edges $E_c$, and indicate how to stitch them when $c > 0$. **(c)** We use an implicit neural representation for the signed distance and for the edge labels, denoted here by the different colors. **(d)** Interpolation in the latent space allows topology changes, here from a sleeveless shirt to a long-sleeve open jacket. The top rows show the front and back panels, the bottom row the reconstructed meshes. **(e)** To parameterize a two-panel garment, we train the implicit network $\mathcal{I}_\Theta$ to predict the signed distance field ($s_f/s_b$) and the edge label field ($c_f/c_b$) that represent the two panels. They are mapped to 3D surfaces by the network $\mathcal{A}_\Phi$.

$c$ of the closest boundary for all points, as shown in Fig. 2(c), which yields a much better behaved minimization problem at training time.

In this paper, we model every garment $G$ using two panels $P^f$ and $P^b$, one for the front and one for the back. $E^f$ and $E^b$ are labels assigned to boundary points. Label 0 denotes unstitched boundary points like collar, waist, or sleeve ends. Labels other than zero denote points in the boundary of one panel that should be stitched to a point with the same label in the other.

In practice, given the database of garments and corresponding 2D panels we generated using publicly available software designed for this purpose [46], we use an *auto-decoding* approach to jointly train $\mathcal{I}_\Theta$ and to associate a latent vector $\mathbf{z}$ to each training garment. To this end, we minimize the loss function obtained by summing over all training panels

$$\mathcal{L}_\mathcal{I} = \sum_{\mathbf{x} \in \Omega} \left| s(\mathbf{x}, \mathbf{z}) - s^{gt}(\mathbf{x}) \right| + \lambda_{CE}\text{CE}(c(\mathbf{x}, \mathbf{z}), c^{gt}(\mathbf{x})) + \lambda_{reg} \|\mathbf{z}\|_2^2 \; , \tag{2}$$

with respect to a separate latent code $\mathbf{z}$ per garment and the network weights $\Theta$ that are used for all garments. CE is the cross-entropy loss, $s^{gt}$ and $c^{gt}$ are the ground-truth signed distance value and the label of the closest seamed edge of $\mathbf{x}$, and $\lambda_{CE}$ and $\lambda_{reg}$ are scalars balancing the influence of the different terms. We handle each garment having a front and a back panel $P^f$ and $P^b$ by two separate network $\mathcal{I}_{\Theta_f}$ and $\mathcal{I}_{\Theta_b}$ with shared latent codes so that it can produce two separate sets of $(s, c)$ values at each $(x_u, x_v)$ location, one for the front and one for the back.

**From 2D panels to 3D Surfaces.** The sewing patterns described above are flat panels whose role is to provide stitching instructions. To turn them into 3D surfaces that can be draped on a human body, we learn a mapping from the 2D panels to the 3D garment draped on the neutral SMPL body. To this end, we introduce a second function, similar to the one used in AtlasNet [48]

$$\mathcal{A}_\Phi : \Omega \times \mathbb{R}^{|\mathbf{z}|} \to \mathbb{R}^3, \; (\mathbf{x}, \mathbf{z}) \mapsto \mathbf{X} \; . \tag{3}$$

It is also implemented by a fully connected network and takes as input a local 2D location $\mathbf{x}$ and a global latent vector $\mathbf{z}$. It returns a 3D position $\mathbf{X}$ for every 2D location $\mathbf{x}$ in the pattern. A key difference with AtlasNet is that we only evaluate $\mathcal{A}_\Phi$ for points $\mathbf{x}$ within the panels, that is points for which the signed distance returned by $\mathcal{I}_\Theta$ of Eq. (1) is not positive. Hence, there is no need to deform or stretch $uv$ patterns. This is in contrast to the square patches of AtlasNet that had to be, which simplifies the training.

Given the latent codes $\mathbf{z}$ learned for each garment in the training database, as described above, we train a separate set of weights $\Phi_f$ and $\Phi_b$ for the front and back of the garments. To this end, we minimize a sum over all front and back training panels of

$$\mathcal{L}_\mathcal{A} = \mathcal{L}_{CHD} + \lambda_n \mathcal{L}_{normal} + \lambda_c \mathcal{L}_{consist} , \tag{4}$$

$$\mathcal{L}_{consist} = \sum_{c>0} \sum_{\mathbf{x} \in E_c} \left\| \mathcal{A}_{\Phi_f}(\mathbf{x}, \mathbf{z}) - \mathcal{A}_{\Phi_b}(\mathbf{x}, \mathbf{z}) \right\|_2^2 . \tag{5}$$

where $\mathcal{L}_{CHD}$, $\mathcal{L}_{normal}$, and $\mathcal{L}_{consist}$ are the Chamfer distance, a normal consistency loss, and a loss term whose minimization ensures that the points on the edges of the front and back panels sewn together—$E_c$ for $c > 0$—are aligned when folding the panels in 3D, $\lambda_n$ and $\lambda_c$ are scalar weights. This assumes that the front and back panels have their seamed edges aligned in 2D, ie. for $c > 0$ and $\mathbf{x} \in \Omega$, if $\mathbf{x}$ has label $c$ on the front panel, then it should also have label $c$ in the back panel, and vice-versa. Our experiments show that $\mathcal{L}_{consist}$ reduces the gap between the front and back panels.

**Meshing and Preserving Differentiability.** $\mathcal{A}_\Phi$ continuously deforms 2D patches that are implicitly defined by $\mathcal{I}_\Theta$. To obtain triangulated meshes from these, we lift a regular triangular 2D mesh defined on $\Omega$ by querying $\mathcal{A}_\Phi$ on each of its vertex, as in [48]. More specifically, we first create a square 2D mesh $\mathbf{T} = \{V_\Omega, F_\Omega\}$ for $\Omega$, where vertices $V_\Omega$ are grid points evenly sampled along orthogonal axis of $\Omega$ and faces $F_\Omega$ are created with Delaunay triangulation. Given the latent code $\mathbf{z}$ of a specific garment, for each vertex $v \in V_\Omega$, we can obtain its signed distance value $s$ and edge label $c$ with $(s, c) = \mathcal{I}_\Theta(v, \mathbf{z})$. We construct the 2D panel mesh $\mathbf{T}_P = \{V_P, F_P\}$ by discarding vertices of $\mathbf{T}$ whose SDF is positive. To get cleaner panel borders, we also keep vertices $v \in V_\Omega$ with $s(v, \mathbf{z}) > 0$ that belong to the mesh edges that cross the 0 iso-level, and adjust their positions to $v - s(v, \mathbf{z}) \nabla s(v, \mathbf{z})$ to project them to the zero level set [49]. The front and back panel 2D meshes are then lifted to 3D by querying $\mathcal{A}_\Phi$ on each vertex. During post-processing, their edges are sewn together to produce the final 3D garment mesh $\mathbf{T}_G(\mathbf{z})$. More details can be found in the Supplementary Material.

$\mathcal{I}_\Theta$ performs as an indicator function to keep vertices with $s \leq 0$, which breaks automatic differentiability. To restore it, we rely on gradients derived in [50, 51] for implicit surface extraction. More formally, assume $v$ is a vertex of the extracted mesh $\mathbf{T}_G$ and $\mathbf{x}$ is the point on the UV space that satisfies $v = \mathcal{A}_\Phi(\mathbf{x}, \mathbf{z})$, then, as proved in the Supplementary Material,

$$\frac{\partial v}{\partial \mathbf{z}} = \frac{\partial \mathcal{A}_\Phi}{\partial \mathbf{z}}(\mathbf{x}, \mathbf{z}) - \frac{\partial \mathcal{A}_\Phi}{\partial \mathbf{x}} \nabla s(\mathbf{x}, \mathbf{z}) \frac{\partial s}{\partial \mathbf{z}}(\mathbf{x}, \mathbf{z}) . \tag{6}$$

Hence, ISP can be used to solve inverse problems using gradient descent, such as fitting garments to partial observations.

## 3.2 Garment Draping

We have defined a mapping from a latent vector $\mathbf{z}$ to a garment draped over the neutral SMPL model [31], which we will refer to as a *rest pose*. We now need to deform the garments to conform to bodies of different shapes and in different poses. To this end, we first train a network that can deform a single garment. We then train a second one that handles the interactions between multiple garment layers and prevents interpenetrations as depicted in Fig. 3.

**Single Layer Draping.** In SMPL, body templates are deformed using LBS. As in [19, 29, 30], we first invoke an extended skinning procedure with the diffused body model formulation to an initial rough estimate of the garment shape. More specifically, given the parameter vectors $\mathbf{B}$ and $\mathbf{\Theta}$ that control the body shape and pose respectively, each vertex $v$ of a garment mesh $\mathbf{T}_G$ is transformed by

$$v_{init} = W(v_{(\mathbf{B},\mathbf{\Theta})}, \mathbf{B}, \mathbf{\Theta}, w(v)\mathcal{W}) , \tag{7}$$

$$v_{(\mathbf{B},\mathbf{\Theta})} = v + w(v)B_s(\mathbf{B}) + w(v)B_p(\mathbf{\Theta}) ,$$

where $W(\cdot)$ is the SMPL skinning function with skinning weights $\mathcal{W} \in \mathbb{R}^{N_B \times 24}$, with $N_B$ being the number of vertices of the SMPL body mesh, and $B_s(\mathbf{B}) \in \mathbb{R}^{N_B \times 3}$ and $B_p(\mathbf{\Theta}) \in \mathbb{R}^{N_B \times 3}$ are the

shape and pose displacements returned by SMPL. $w(v) \in \mathbb{R}^{N_B}$ are diffused weights returned by a neural network, which generalizes the SMPL skinning to any point in 3D space. The training of $w(v)$ is achieved by smoothly diffusing the surface values, as in [29].

This yields an initial deformation, such that the garment roughly fits the underlying body. A displacement network $\mathcal{D}_s$ is then used to refine it. $\mathcal{D}_s$ is conditioned on the body shape and pose parameters $\mathbf{B}, \mathbf{\Theta}$ and on the garment specific latent code $\mathbf{z}$. It returns a displacement map $D_s = \mathcal{D}_s(\mathbf{B}, \mathbf{\Theta}, \mathbf{z})$ in UV space. Hence, the vertex $v_{init}$ with UV coordinates $(x_u, x_v)$ in $\Omega$ is displaced accordingly and becomes $\tilde{v} = v_{init} + D_s[x_u, x_v]$, where $[\cdot, \cdot]$ denotes standard array addressing. $\mathcal{D}_s$ is implemented as a multi-layer perceptron (MLP) that outputs a vector of size $6N^2$, where $N$ is the resolution of UV mesh $\mathbf{T}$. The output is reshaped to two $N \times N \times 3$ arrays to produce the front and back displacement maps.

To learn the deformation for various garments without collecting any ground-truth data and train $\mathcal{D}_s$ in a self-supervised fashion, we minimize the physics-based loss from [18]

$$\mathcal{L}_{phy} = \mathcal{L}_{strain} + \mathcal{L}_{bend} + \mathcal{L}_{gravity} + \mathcal{L}_{BGcol} , \tag{8}$$

where $\mathcal{L}_{strain}$ is the membrane strain energy caused by the deformation, $\mathcal{L}_{bend}$ the bending energy raised from the folding of adjacent faces, $\mathcal{L}_{gravity}$ the gravitational potential energy, and $\mathcal{L}_{BGcol}$ the penalty for body-garment collisions.

**Multi-Layer Draping.** When draping independently multiple garments worn by the same person using the network $\mathcal{D}_s$ introduced above, the draped garments can intersect, which is physically impossible and must be prevented. We now show that our ISP model makes that straightforward first in the case of two overlapping garments, and then in the case of arbitrarily many.

Consider an outer garment $\mathbf{T}_G^o$ and an inner garment $\mathbf{T}_G^i$ with rest state vertices $V_o$ and $V_i$. We first drape them independently on a target SMPL body with $\mathcal{D}_s$, as described for single layer draping. This yields the deformed outer and underlying garments with vertices $\tilde{V}_o$ and $\tilde{V}_i$, respectively. We then rely on a second network $\mathcal{D}_m$ to predict corrective displacements, conditioned on the outer garment geometry and repulsive virtual forces produced by the intersections with the inner garment.

In ISP, garments are represented by mapping 2D panels to 3D surfaces. Hence their geometry can be stored on regular 2D grids on which a convolutional network can act. In practice, we first encode the *rest state* of the outer garment $\mathbf{T}_G^o$ into a 2D position map $M_r$. The grid $M_r$ records the 3D location of the vertex $v_o \in V_o$ at its $(x_u, x_v)$ coordinate as $M_r[x_u, x_v] = v_o$ within the panel boundaries, $M_r[x_u, x_v] = (0, 0, 0)$ elsewhere. Concatenating both front and back panels yields a $N \times N \times 6$ array, that is, a 2D array of spatial dimension $N$ with 6 channels. After draping, the same process is applied to encode the geometry of $\mathbf{T}_G^o$ into position map $M_d$, using vertices $\tilde{V}_o$ instead of $V_o$ this time. Finally, for each vertex $\tilde{v}_o \in \tilde{V}_o$, we take the repulsive force acting on it to be

$$f(\tilde{v}_o) = max(0, (\tilde{v}_i - \tilde{v}_o) \cdot \mathbf{n}_i)\mathbf{n}_i , \tag{9}$$

where $\tilde{v}_i$ is the closest vertex in $\tilde{V}_i$, $\mathbf{n}_i$ is the normal of $\tilde{v}_i$, and $\cdot$ represents the dot product. The repulsive force is also recorded in the UV space to generate the 2D force map $M_f$. Note that it is 0 for vertices that are already outside of the inner garment, for which no collision occurs. Given the forces $M_f$, the garment geometry in the rest state $M_r$ and after draping $M_d$, the network $\mathcal{D}_m$ predicts a vertex displacements map $D_m = \mathcal{D}_m(M_r, M_d, M_f)$ for the outer garment to resolve intersections, as shown in Fig. 3. We replace the vertex $\tilde{v}_o \in \tilde{V}_o$ with coordinates $(x_u, x_v)$ by $\tilde{v}_o^* = \tilde{v}_o + D_m[x_u, x_v]$. $\mathcal{D}_m$ is implemented as a CNN, and it can capture the local geometry and force information of vertices from the input 2D maps. The training of $\mathcal{D}_m$ is self-supervised by

$$\mathcal{L}_{\mathcal{D}_m} = \mathcal{L}_{phy} + \lambda_g \mathcal{L}_{GGcol} + \lambda_r \mathcal{L}_{reg} , \tag{10}$$

$$\mathcal{L}_{GGcol} = \sum_{\tilde{v}_o^*} max(0, \epsilon - (\tilde{v}_o^* - \tilde{v}_i) \cdot \mathbf{n}_i)^3 , \text{ and } \mathcal{L}_{reg} = \|\mathcal{D}_m(M_r, M_d, M_f)\|_2^2 ,$$

where $\mathcal{L}_{phy}$ is the physics-based loss of Eq. (8), $\lambda_g$ and $\lambda_r$ are weighting constants, and $\epsilon$ is a safety margin to avoid collisions. $\mathcal{L}_{GGcol}$ penalizes the intersections between the outer and underlying garments, and $\mathcal{L}_{reg}$ is an $L_2$ regularization on the output of $\mathcal{D}_m$.

Given a pair of garments, our layering network $\mathcal{D}_m$ only deforms the outer garment to resolve collisions, leaving the inner one untouched. Given more than 2 overlapping garments, the process can be iterated to layer them in any desired order over a body, as detailed in the Supplementary Material.

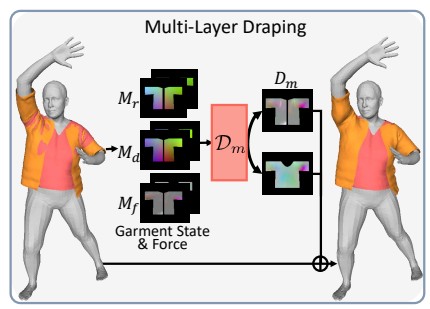

Figure 3: **Multi-layer Draping**. The network $\mathcal{D}_m$ uses the garment geometry and forces encoded in the input UV maps to resolve garment intersections for multi-layered draping.

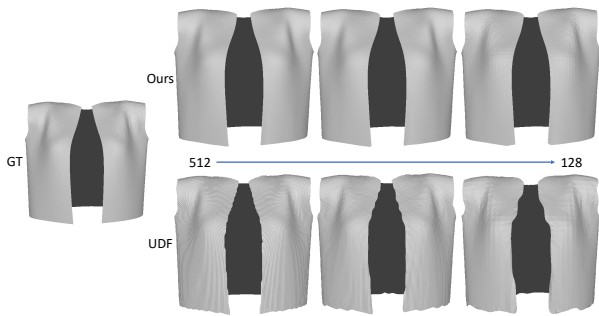

Figure 4: **Comparison between UDF and our method in resolution 512, 256 and 128**. Our reconstructions (top row) are more accurate than those of UDF (bottom row) which have artifacts and uneven borders.

## 4 Experiments

### 4.1 Dataset, Evaluation Metrics, and Baseline

To create training and test sets, we used the software of [46] to generate sewing patterns and the corresponding 3D garment meshes in their rest state, that is draped over a T-Posed body. Our training set comprises 400 shirts, 300 skirts, and 200 pairs of trousers. For testing, we use 20 shirts, 20 skirts, and 20 pairs of trousers. As discussed in Section 3.2, we trained the draping models using SMPL body poses $\Theta$ randomly sampled from the AMASS [52] dataset and body shapes $\mathbf{B}$ uniformly sampled from $[-2, 2]^{10}$. We use the Chamfer Distance (CHD) and Normal Consistency (NC) to quantify garment reconstruction accuracy, along with the percentage of the garment mesh area that undergoes interpenetrations (Intersection), as in [30].

We compare our approach against recent and state-of-the-art methods DIG [29] and DrapeNet [30], which, like ours, can drape various garments over bodies of different shapes in arbitrary poses. Like our ISPs, DrapeNet is self-supervised using a physics-based loss and can prevent unwanted intersections between top (shirts) and bottom (trousers) garments. By contrast, DIG is a fully supervised method and cannot handle garment intersections.

### 4.2 Garment Reconstruction

| Train | CHD ($\times 10^{-4}$, $\downarrow$) | NC (%, $\uparrow$) | Time (ms, $\downarrow$) | Test | CHD ($\times 10^{-4}$, $\downarrow$) | NC (%, $\uparrow$) |
|---|---|---|---|---|---|---|
| UDF - 128 | 0.641 | 98.86 | 601 | UDF - 128 | 0.803 | 97.71 |
| UDF - 256 | 0.338 | 99.20 | 2379 | UDF - 256 | 0.493 | 98.44 |
| UDF - 512 | 0.262 | 98.77 | 13258 | UDF - 512 | 0.424 | 98.09 |
| Ours - 128 | 0.454 | 99.20 | **25** | Ours - 128 | 0.579 | 98.70 |
| Ours - 256 | 0.290 | 99.37 | 77 | Ours - 256 | 0.392 | 98.83 |
| Ours - 512 | **0.250** | **99.41** | 261 | Ours - 512 | **0.349** | **98.89** |

Table 1: Comparison of our method to UDF on shirts under the resolutions of 128, 256 and 512.

We first consider 3D garments in their rest pose and compare the accuracy of our ISPs against that of the UDF-based representation [30]. In Fig. 4, we provide qualitative results in a specific case and for resolutions of Marching Cubes [53] ranging from 512 to 128. Our result is visually superior and more faithful to the ground-truth garment, with UDF producing more artifacts and uneven borders, especially at resolution 512. The quantitative results reported in Tab. 1 for the training and test sets confirm this. For the test set garments, we reconstruct them by optimizing the latent code to minimize the Chamfer distance between the predicted and ground truth meshes, using the gradients of Eq. (6). Our approach consistently outperforms UDF at all resolutions while being faster, mostly because we evaluate our network on a 2D implicit field while UDF does it on a 3D one. For example, at resolution 256, our reconstruction time is 77 ms on an Nvidia V100 GPU, while UDF requires 2379 ms. Interestingly, increasing the resolution from 256 to 512 improves the reconstruction accuracy

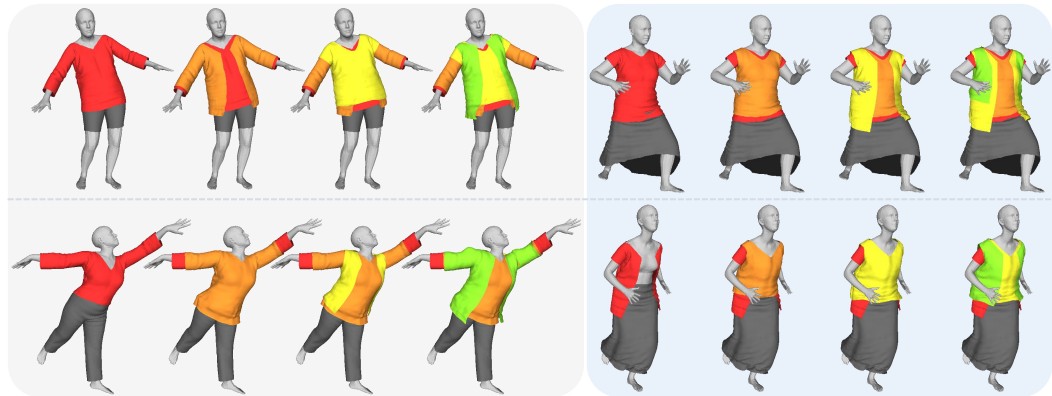

Figure 5: **Realistic garment draping** with our method, for different combinations of garments.

of our method, whereas that of UDF's drops. This is because precisely learning the 0 iso-surface of a 3D UDF is challenging, resulting in potentially inaccurate normals near the surface [54, 47]. We present similar results for trousers and skirts in the supplementary material.

## 4.3 Garment Draping

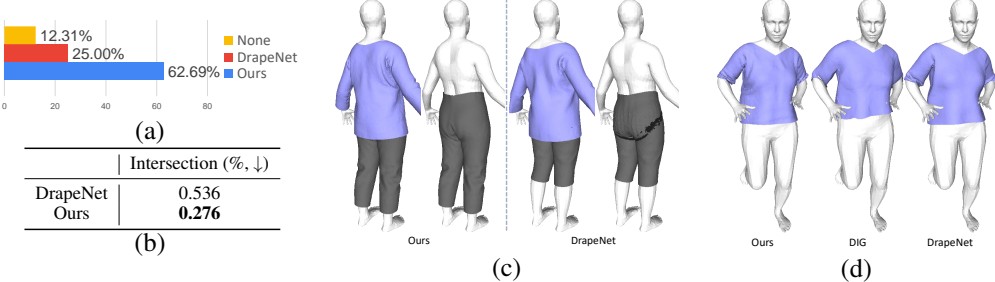

|  | Intersection (%, ↓) |
|---|---|
| DrapeNet | 0.536 |
| Ours | **0.276** |

(b)

Figure 6: **Draping evaluation.** (a) Human evaluation results. None refers to no preference. (b) Percentage of intersecting areas. (c) For each method, left is the draping results for a shirt and a pair of trousers, and right with only the pants. (d) The draping results of our method, DIG and DrapeNet.

Figs. 1 and 5 showcase our method's ability to realistically drape multiple garments with diverse geometry and topology over the body. Our approach can handle multi-layered garments, which is not achievable with DrapeNet. Additionally, unlike the method of [38] that is limited to multi-layered garments on the T-posed body, our method can be applied to bodies in arbitrary poses.

As pointed out in [13, 30], there are no objective metrics for evaluating the realism of a draping. Therefore, we conducted a human evaluation. As in [30], we designed a website displaying side-by-side draping results generated by our method and DrapeNet for the same shirts and trousers, on the same bodies. Participants were asked to select the option that seemed visually better to them, with the third option being "I cannot decide". 64 participants took part in the study, providing a total of 884 responses. As shown in Fig. 6(a), the majority of participants preferred our method (62.69% vs. 25.00%), demonstrating the higher fidelity of our results. The lower intersection ratio reported in Fig. 6(b) further demonstrates the efficacy of our draping model. In Figs. 6(c) and (d), we compare qualitatively our method against DrapeNet and DIG.

## 4.4 Recovering Multi-Layered Garments from Images

Thanks to their differentiability, our ISPs can be used to recover multi-layered garments from image data, such as 2D garment segmentation masks. Given an image of a clothed person, we can obtain the estimation of SMPL body parameters $(\mathbf{B}, \mathbf{\Theta})$ and the segmentation mask $\mathbf{S}$ using the algorithms of [56, 57]. To each detected garment, we associate a latent code $\mathbf{z}$ and reconstruct garment meshes

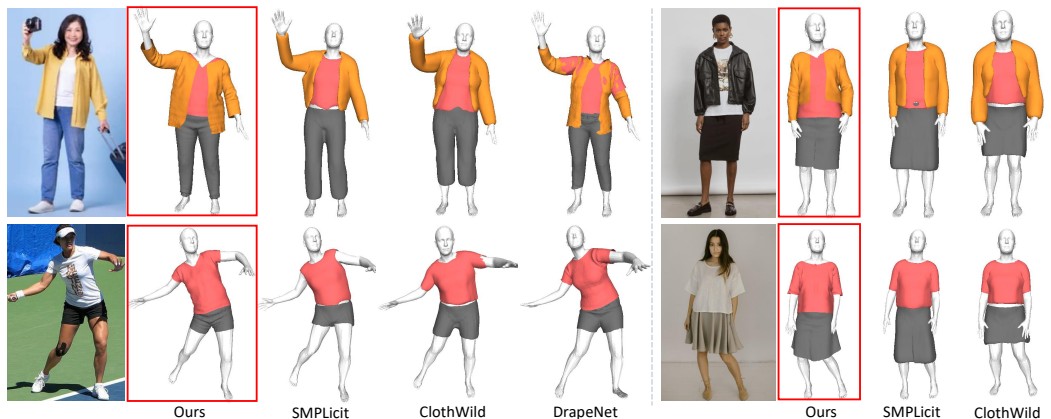

Figure 7: **Garment recovery from images**. We compare the meshes recovered by our method and the state of the art methods SMPLicit [28], ClothWild [55], DrapeNet [30] (unavailable for skirts).

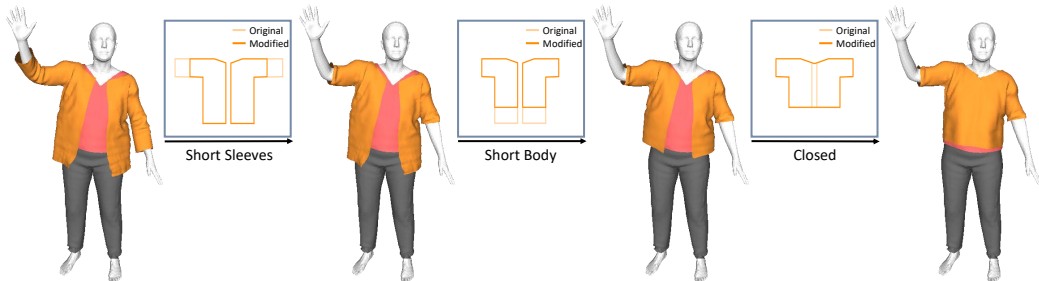

Figure 8: **Shape editing**. Garment attributes can be edited by modifying the sewing pattern.

by minimizing

$$\mathbf{z}_{1:N}^{*} = \underset{\mathbf{z}_{1:N}}{\mathrm{argmin}}\, L_{\mathrm{IoU}}(R(\mathcal{G}(\mathbf{B},\mathbf{\Theta},\mathbf{z}_{1:N}) \oplus \mathcal{M}(\mathbf{B},\mathbf{\Theta})),\mathbf{S})\,, \tag{11}$$

$$\mathcal{G}(\mathbf{B},\mathbf{\Theta},\mathbf{z}_{1:N}) = \mathcal{D}(\mathbf{B},\mathbf{\Theta},\mathbf{z}_1,\mathbf{T}_G(\mathbf{z}_1)) \oplus \cdots \oplus \mathcal{D}(\mathbf{B},\mathbf{\Theta},\mathbf{z}_N,\mathbf{T}_G(\mathbf{z}_N)),$$

where $L_{\mathrm{IoU}}$ is the IoU loss [58] over the rendered and the given mask, $R(\cdot)$ is a differentiable renderer [59], $N$ is the number of detected garments, and $\oplus$ represents the operation of mesh concatenation. $\mathcal{M}$ is the SMPL body mesh, while $\mathcal{G}(\mathbf{B},\mathbf{\Theta},\mathbf{z}_{1:N})$ is the concatenation of garment meshes reconstructed from the implicit sewing pattern as $\mathbf{T}_G(\mathbf{z}_i)$ and then draped by our draping model $\mathcal{D} = \mathcal{D}_m \circ \mathcal{D}_s$. In practice, the minimization is performed from the outermost garment to the innermost, one by one. Further details can be found in the Supplementary Material.

Fig. 7 depicts the results of this minimization. Our method outperforms the state-of-the-art methods SMPLicit [28], ClothWild [55] and DrapeNet [30], given the same garment masks. The garments we recover exhibit higher fidelity and have no collisions between them or with the underlying body.

### 4.5  Garment Editing

As the panel shape of the sewing pattern determines the shape of the garment, we can easily manipulate the garment mesh by editing the panel. Fig. 8 depicts this editing process. We begin by moving the sleeve edges of the panel inwards to shorten the sleeves of the mesh, then move the bottom edges up to shorten the length of the jacket, and finally remove the edges for the opening to close it. This is achieved by minimizing

$$L(\mathbf{z}) = d(E(\mathbf{z}),\tilde{E})\,, \text{ where } E(\mathbf{z}) = \{\mathbf{x}|s(\mathbf{x},\mathbf{z}) = 0, \mathbf{x} \in \Omega\}\,, \tag{12}$$

w.r.t. $\mathbf{z}$. $d(\cdot)$ computes the chamfer distance, $\tilde{E}$ represents the edges of the modified panel, and $E(\mathbf{z})$ represents the 0 iso-level extracted from $\mathcal{I}_{\Theta}$, that is, the edges of the reconstructed panel. Our sewing pattern representation makes it easy to specify new edges by drawing and erasing lines in

the 2D panel images, whereas the fully implicit garment representation of [30] requires an auxiliary classifier to identify directions in the latent space for each garment modification. As shown in the supplementary material, the texture of the garment can also be edited by drawing on the UV panels.

## 5 Conclusion

We have introduced a novel representation for garments to be draped on human bodies. The garments are made of flat 2D panels whose boundary is defined by a 2D SDF. To each panel is associated a 3D surface parameterized by the 2D panel coordinates. Hence, different articles of clothing are represented in a standardized way. This allows the draping of multi-layer clothing on bodies and the recovery of such clothing from single images. Our current implementation assumes quasi-static garments and only deforms the outer garment to solve collisions in multi-layer draping. In future work, we will introduce garment dynamics and focus on more accurate physical interactions between the outer and inner garments.

**Acknowledgement.** This project was supported in part by the Swiss National Science Foundation.

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

# Appendix

# A  Additional Results

## A.1  Texture Editing

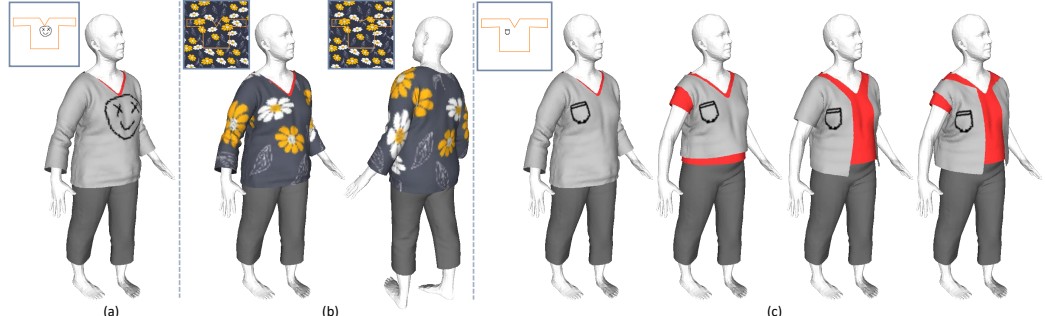

Figure 9: **Texture editing. (a)** A smiling face drawn on the front panel. **(b)** A flower pattern painted on both the front and the back panels. **(c)** A pocket drawn for the left shirt can be transferred to different garments.

As illustrated in Fig. 9, with our ISP, the texture of the garment can be easily edited by drawing on the UV panels. The figures drawn on the panel of one garment can be directly transferred to others as shown by Fig. 9(c), since panels are defined on the same UV space.

## A.2  Garment Reconstruction

### A.2.1  Evaluation Results

In Tabs. 2 and 3 we report the reconstruction results for skirts and trousers on the training and the test set. Similar to the results on shirts shown in the main paper, our method achieves better reconstruction quality than UDF [30] with lower CHD and higher NC at all resolutions, and needs less time to

| Train | CHD ($\times 10^{-4}$, ↓) | NC (%, ↑) | Time (ms, ↓) | Test | CHD ($\times 10^{-4}$, ↓) | NC (%, ↑) |
|---|---|---|---|---|---|---|
| UDF - 128 | 0.714 | 98.53 | 690 | UDF - 128 | 0.734 | 97.79 |
| UDF - 256 | 0.408 | 98.93 | 2784 | UDF - 256 | 0.403 | 98.64 |
| UDF - 512 | 0.331 | 98.43 | 17362 | UDF - 512 | 0.324 | 98.27 |
| Ours - 128 | 0.538 | 98.80 | **25** | Ours - 128 | 0.583 | 98.62 |
| Ours - 256 | 0.321 | 99.08 | 75 | Ours - 256 | 0.362 | 98.85 |
| Ours - 512 | **0.267** | **99.14** | 262 | Ours - 512 | **0.304** | **98.89** |

Table 2: Comparison of our method to UDF on skirts under the resolutions of 128, 256 and 512.

| Train | CHD ($\times 10^{-4}$, ↓) | NC (%, ↑) | Time (ms, ↓) | Test | CHD ($\times 10^{-4}$, ↓) | NC (%, ↑) |
|---|---|---|---|---|---|---|
| UDF - 128 | 0.758 | 98.22 | 661 | UDF - 128 | 0.752 | 97.41 |
| UDF - 256 | 0.430 | 98.56 | 2650 | UDF - 256 | 0.425 | 98.09 |
| UDF - 512 | 0.342 | 98.33 | 17141 | UDF - 512 | 0.350 | 97.63 |
| Ours - 128 | 0.545 | 98.28 | **25** | Ours - 128 | 0.529 | 98.03 |
| Ours - 256 | 0.363 | 98.59 | 82 | Ours - 256 | 0.346 | 98.31 |
| Ours - 512 | **0.317** | **98.62** | 269 | Ours - 512 | **0.300** | **98.32** |

Table 3: Comparison of our method to UDF on trousers under the resolutions of 128, 256 and 512.

reconstruct a single mesh. Figs. 10 to 12 show the qualitative results reconstructed by our method for shirts, skirts and trousers respectively.

### A.2.2 Latent Space Interpolation

In Figs. 13 to 15, we display the results of interpolation in the latent space of shirts, skirts and trousers respectively. We observe a smooth transformation in both the reconstructed sewing patterns and the garment meshes, despite the different topology and geometry of the given garments.

### A.2.3 Comparison with AtlasNet

In Fig. 16, we compare our method with AtlasNet [48] which learns to deform a square patch. AtlasNet struggles to learn a mapping function capable of accurately deforming the square patch to produce a surface that matches the ground truth, especially in the collar region. In contrast, our method leverages the pattern parameterization network $\mathcal{I}_\Theta$ to simplify the training of our mapping function $\mathcal{A}_\Phi$. Specifically, our approach only requires learning the mapping for points within the panels, resulting in a reconstruction that is more faithful to the ground truth.

### A.2.4 Ablation Study

To investigate the impact of the loss terms of Eq. (4) in the main paper on reconstruction quality, we performed an ablation study. We report the results of reconstructing 300 skirts using models trained with and without $\mathcal{L}_{normal}$ in Tab. 4. We observe that training with $\mathcal{L}_{normal}$ reduces the CHD and increases the NC, thus improving the reconstruction accuracy. In contrast, without $\mathcal{L}_{normal}$, the

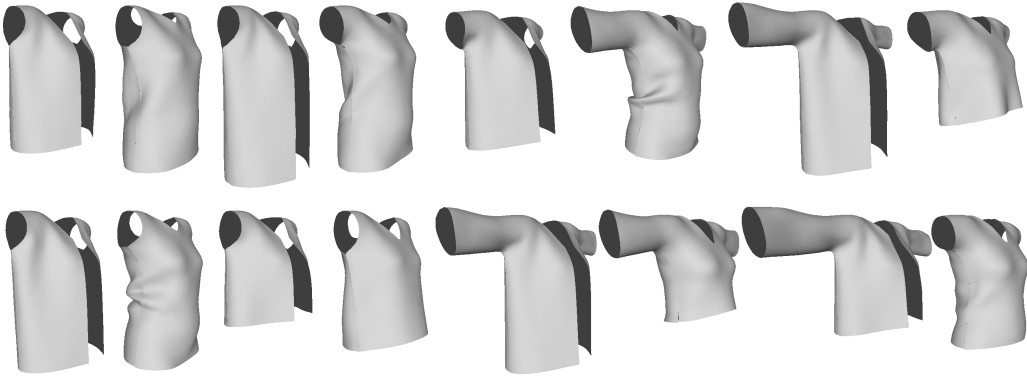

Figure 10: Reconstruction samples of shirts.

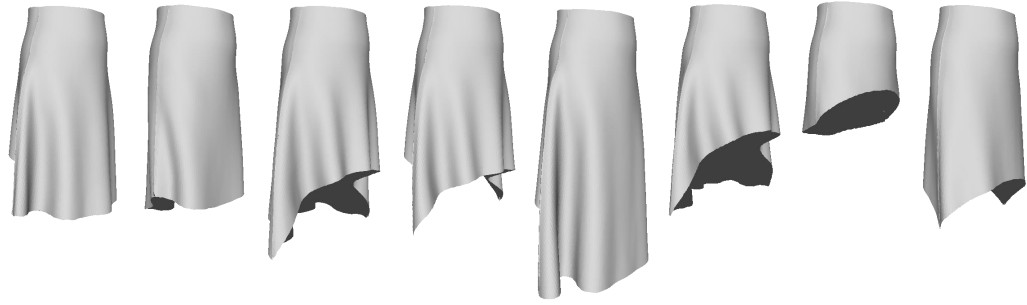

Figure 11: Reconstruction samples of skirts.

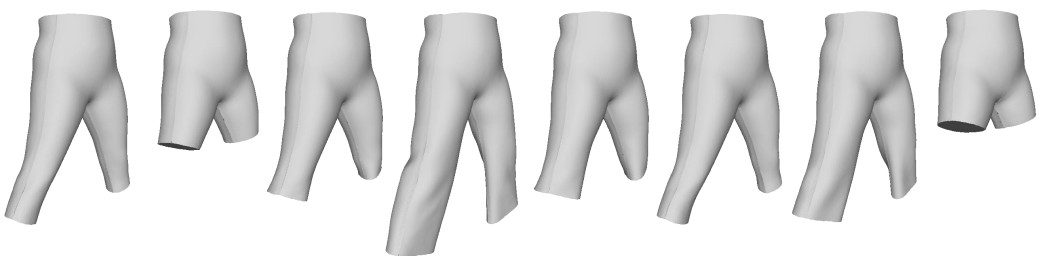

Figure 12: Reconstruction samples of trousers.

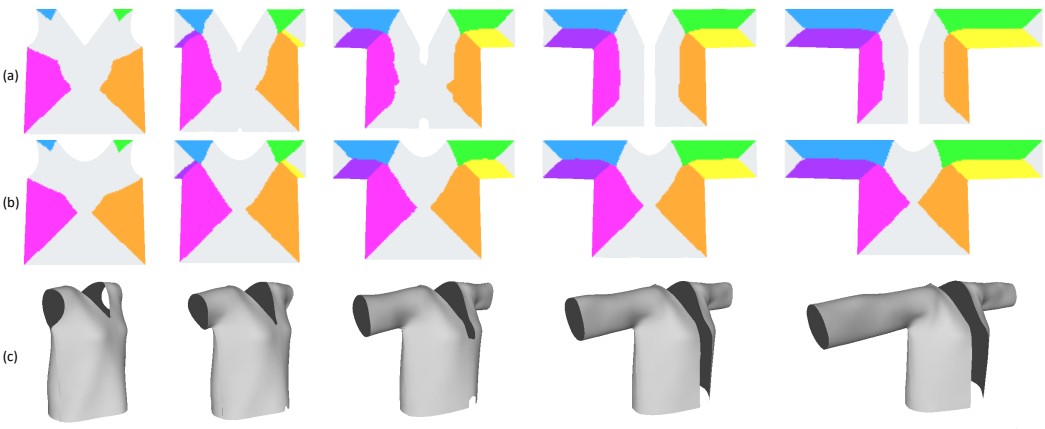

Figure 13: **Interpolation**. We interpolate the latent code from a sleeveless shirt to a long-sleeve jacket. (a) and (b) show the reconstructed front and back panels, where the colors on them denote the edge label fields. (c) shows the reconstructed mesh.

|  | CHD ($\times 10^{-4}$, $\downarrow$) | NC (%, $\uparrow$) |
|---|---|---|
| w/o $\mathcal{L}_{normal}$ | 0.324 | 98.69 |
| w/ $\mathcal{L}_{normal}$ | 0.321 | 99.08 |

Table 4: Comparison of the results reconstructed by the models trained w/o and w/ $\mathcal{L}_{normal}$.

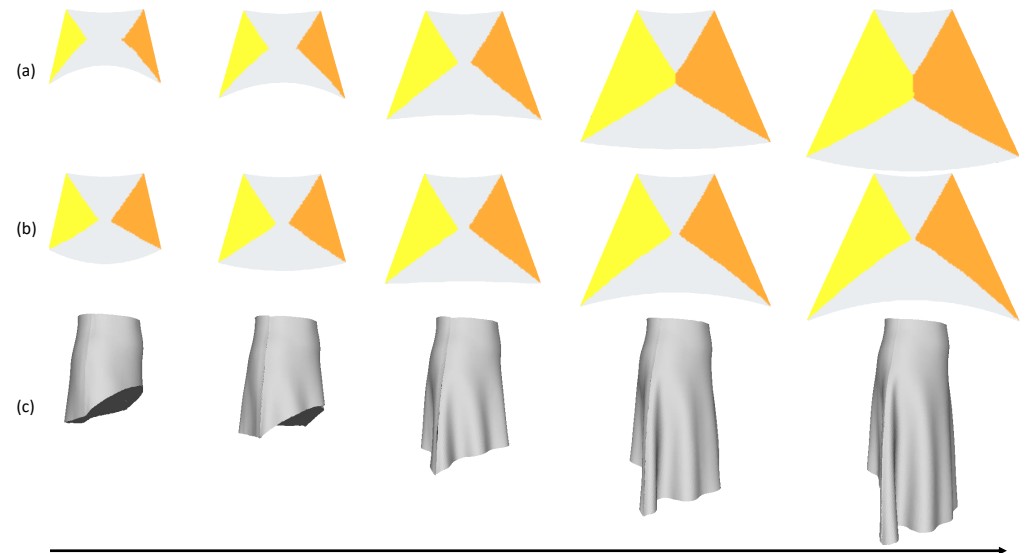

Figure 14: **Interpolation**. We interpolate the latent code from a short tight skirt to a long loose skirt. (a) and (b) show the reconstructed front and back panels, where the colors on them denote the edge label fields. (c) shows the reconstructed mesh.

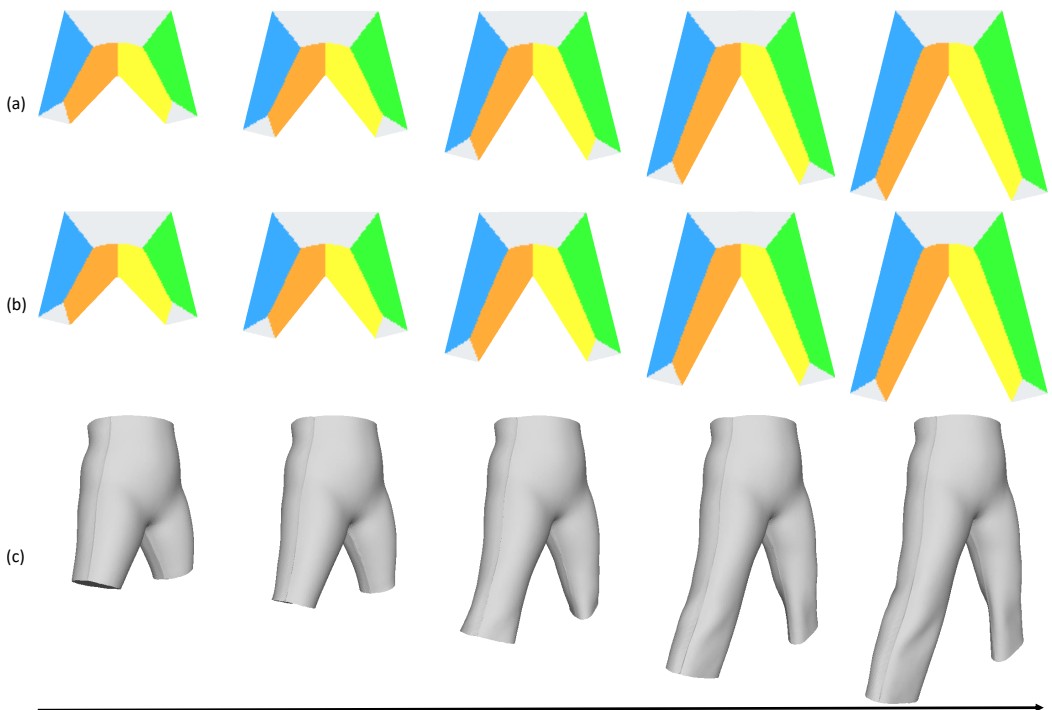

Figure 15: **Interpolation**. We interpolate the latent code from a pair of short trousers to a pair of long trousers. (a) and (b) show the reconstructed front and back panels, where the colors on them denote the edge label fields. (c) shows the reconstructed mesh.

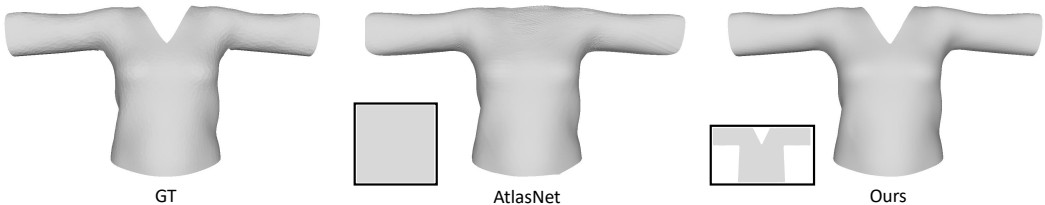

GT  AtlasNet  Ours

Figure 16: **Comparison with AtlasNet**. The meshes in black boxes are the source patches where the mapping function is applied: a square patch for AtlastNet, a garment panel for our method.

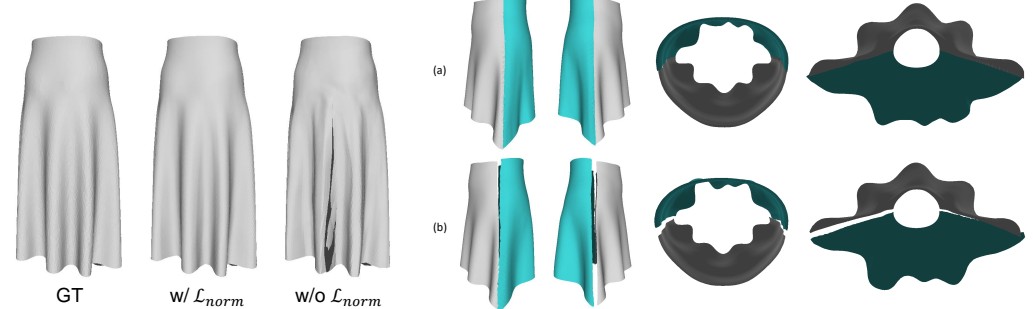

GT  w/ $\mathcal{L}_{norm}$  w/o $\mathcal{L}_{norm}$  (a)  (b)

Figure 17: **Ablation on** $\mathcal{L}_{normal}$. Training without it yields flipped triangle faces (shown in dark grey).

Figure 18: **Ablation on** $\mathcal{L}_{consist}$. The skirts reconstructed by the models trained (a) with $\mathcal{L}_{consist}$ and (b) without $\mathcal{L}_{consist}$. The latter has a gap between its front and back panels.

model fails to learn the correct parameterization for the garment, resulting in a mesh with inverted faces as illustrated in Fig. 17. Fig. 18 shows a comparison of the results obtained by training models without and with $\mathcal{L}_{consist}$ to help stitch the front and back panels. We can notice that without $\mathcal{L}_{consist}$, a spatial gap exists between the front (in gray) and back (in cyan) surfaces.

### A.2.5 Vizualization of the Chamfer Distance

A visualization of the spatial error distribution can be found in Fig. 19. In this figure, we compare the error distribution between our reconstructions and those produced by UDFs. Our reconstructions exhibit lower error across the entire surface compared to UDFs.

### A.3 Garment Draping

Fig. 20 and Fig. 21(a) present additional comparisons of draping results for our method, DIG [29] and DrapeNet [30]. Our results show higher fidelity and fewer artifacts compared to the other methods. We also compare our method with SNUG [18], a self-supervised method that relies on mesh templates for garment representation and trains one network for each clothing item. In Fig. 21(b), we show the qualitative results on the same shirt. Our results are either comparative or visually superior to those of SNUG, despite using a single draping network for a whole garment category.

### A.4 Multi-Layer Draping

To assess the effectiveness of our multi-layering model, we conducted an experiment to measure the intersection ratio between different layers of garments, and between the garments and the body. We used the 673 unseen poses from the test set of AMASS dataset [52] and 673 shape parameters uniformly sampled from the continuous space of $[-2, 2]^{10}$ to generate 673 unique unseen bodies. For each body, we randomly generated five unseen garments (one skirt or one pair of trousers, with four shirts) by interpolating the garment latent codes and draped them on the body. The first layer contained the skirt or trousers, while the shirts were on the following layers.

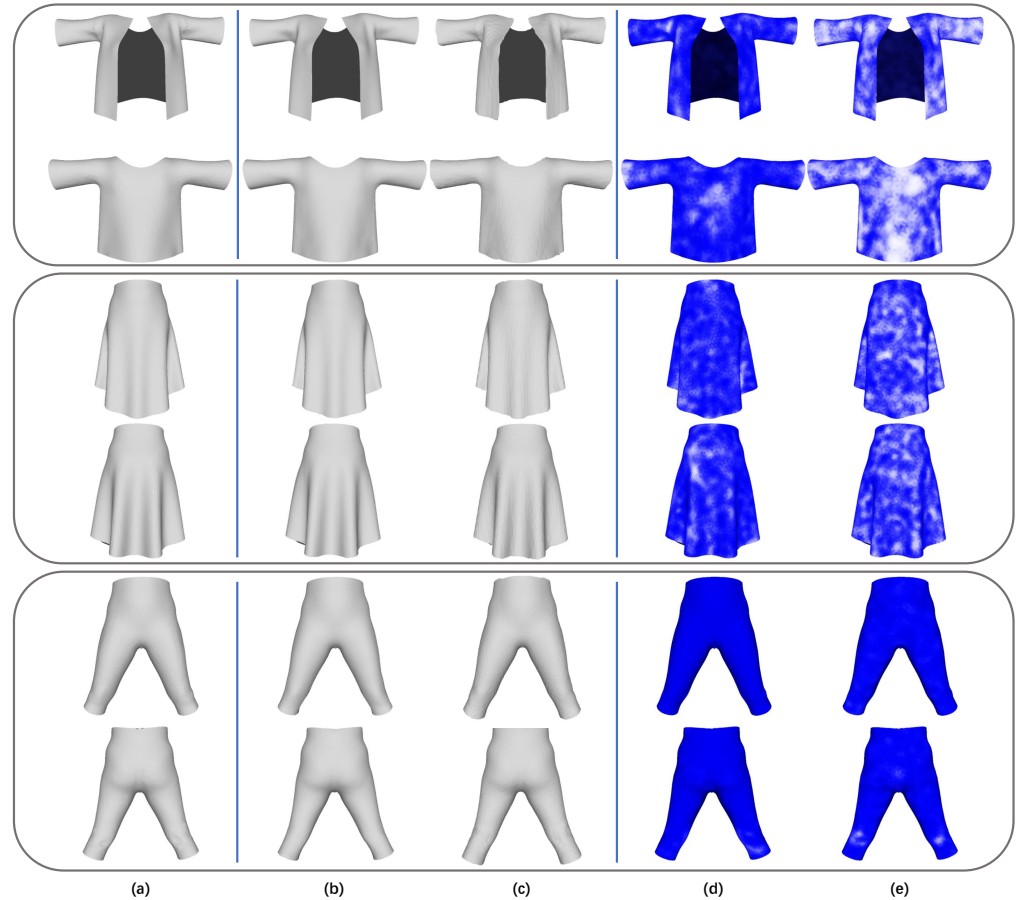

|     |     |     |     |     |
| --- | --- | --- | --- | --- |
| (a) | (b) | (c) | (d) | (e) |

Figure 19: **Comparative analysis of mesh reconstruction and spatial error distribution for shirt, skirt and trousers**. (a) represents the ground truth mesh. (b) and (d) illustrate the mesh reconstruction and corresponding spatial error distribution for our proposed method. (c) and (e) display the same for UDFs. Each row provides both front (upper half) and back (lower half) views. In (d) and (e), error magnitude is indicated by color gradation, with white representing large errors and blue small errors.

The corresponding results are shown in Tabs. 5 and 6, where the diagonal values represent the intersection ratio between the body and the $i$-th ($i = 1, 2, 3, 4, 5$) layer, and the other values represent the intersection between different layers of garments. It is important to note that we cannot fairly compare these numbers to any other method, as they do not support layering. The left and right sides of the tables correspond to the results obtained with and without the layering procedure, respectively. As shown in the tables, the significantly lower intersection ratios on the left side demonstrate the effectiveness of our model in handling intersections with the body and between different garments. Another noteworthy observation is that the intersections tend to increase as we layer more garments. This behavior is explained in Appendix B.4, where we clarify that our multi-layer draping model $\mathcal{D}_m$ is trained solely on garments obtained through single-layer draping, which are relatively close to the body. However, it is worth mentioning that even after five layers, the intersections still remain at a low level ($\sim 2\%$).

# B  Technical Details

## B.1  Sewing Patterns for Trousers and Skirts

In Fig. 22, we show the sewing patterns used in our experiments for trousers and skirts.

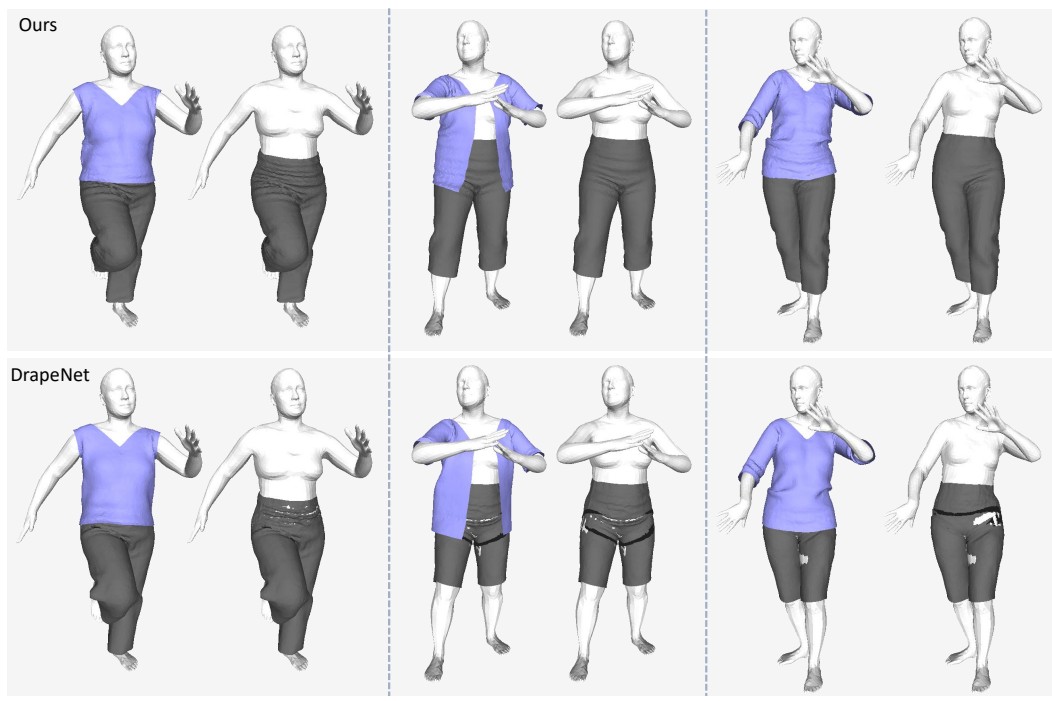

Figure 20: The comparison of draping results of our method and DrapeNet.

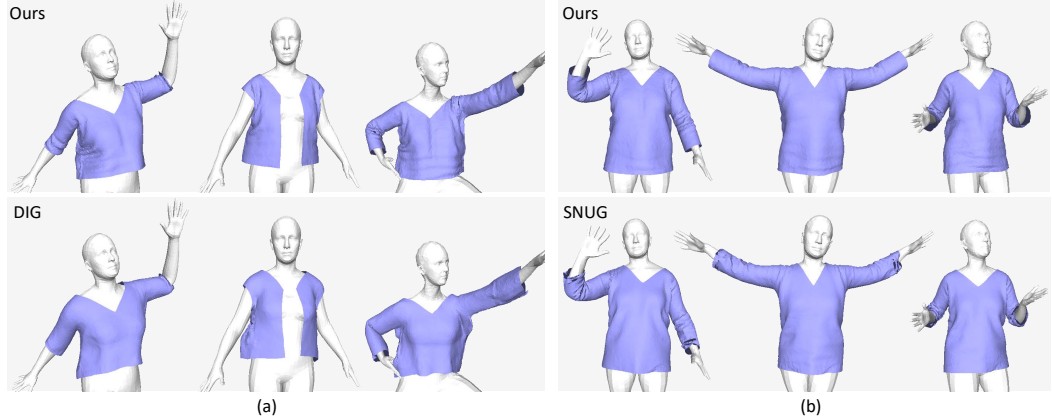

Figure 21: The comparison of draping results for (a) our method vs. DIG, and (b) our method vs. SNUG.

| w/ $\mathcal{D}_m$ | 1 | 2 | 3 | 4 | 5 | w/o $\mathcal{D}_m$ | 1 | 2 | 3 | 4 | 5 |
|---|---|---|---|---|---|---|---|---|---|---|---|
| 1 | 0.000 | 0.067 | 0.001 | 0.000 | 0.000 | 1 | 0.000 | 0.456 | 0.340 | 0.357 | 0.233 |
| 2 | - | 0.003 | 0.633 | 0.608 | 0.636 | 2 | - | 0.004 | 28.033 | 26.378 | 26.246 |
| 3 | - | - | 0.028 | 1.224 | 1.158 | 3 | - | - | 0.046 | 24.729 | 24.679 |
| 4 | - | - | - | 0.019 | 2.449 | 4 | - | - | - | 0.073 | 27.821 |
| 5 | - | - | - | - | 0.035 | 5 | - | - | - | - | 0.052 |

Table 5: Evaluation of intersections with the underlying body and between garments on different layers (the 1st layer is a **skirt**). Left and right are the results of draping with and without the multi-layering network $\mathcal{D}_m$, respectively. The unit is %.

| w/ $\mathcal{D}_m$ | 1 | 2 | 3 | 4 | 5 | w/o $\mathcal{D}_m$ | 1 | 2 | 3 | 4 | 5 |
|---|---|---|---|---|---|---|---|---|---|---|---|
| 1 | 0.185 | 0.048 | 0.000 | 0.000 | 0.000 | 1 | 0.185 | 0.225 | 0.259 | 0.223 | 0.134 |
| 2 | - | 0.003 | 0.631 | 0.588 | 0.637 | 2 | - | 0.004 | 28.033 | 26.378 | 26.246 |
| 3 | - | - | 0.030 | 1.227 | 1.161 | 3 | - | - | 0.046 | 24.729 | 24.679 |
| 4 | - | - | - | 0.020 | 2.426 | 4 | - | - | - | 0.073 | 27.821 |
| 5 | - | - | - | - | 0.028 | 5 | - | - | - | - | 0.052 |

Table 6: Evaluation of intersections with the underlying body and between garments on different layers (the 1st layer is a pair of **trousers**). Left and right are the results of draping with and without the multi-layering network $\mathcal{D}_m$, respectively. The unit is %.

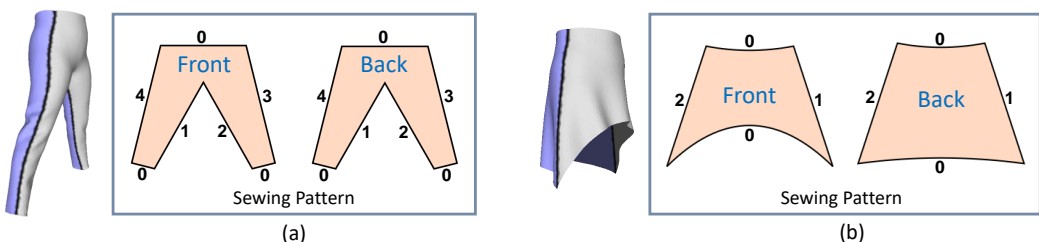

Figure 22: The sewing patterns for (a) trousers and (b) skirts. The front mesh surfaces are in gray and the back ones in blue.

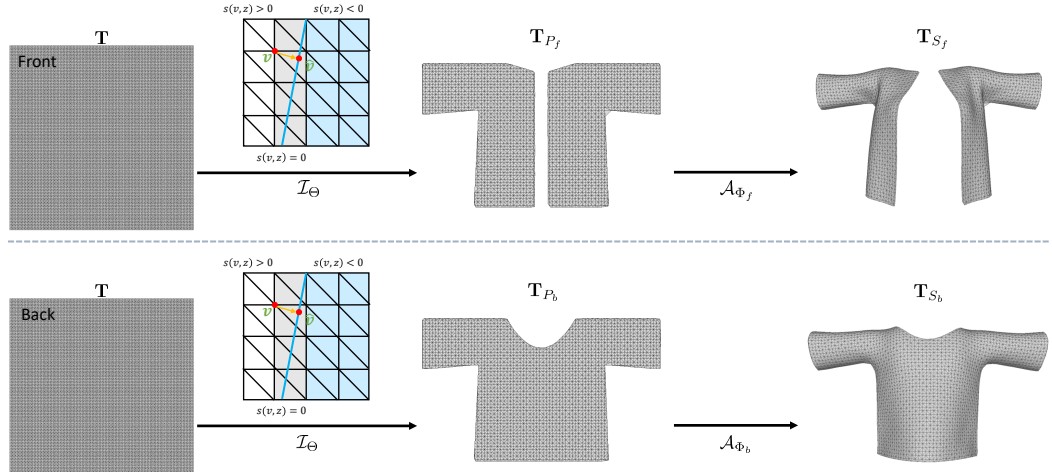

Figure 23: **Meshing Process.** Starting with a square 2D mesh $\mathbf{T}$, we first extract the front and back panel meshes $\mathbf{T}_{P_f}$ and $\mathbf{T}_{P_b}$ using the implicit function $\mathcal{I}_\Theta$. Then we lift $\mathbf{T}_{P_f}$ and $\mathbf{T}_{P_b}$ to 3D to get the surface meshes $\mathbf{T}_{S_f}$ and $\mathbf{T}_{S_b}$ by querying $\mathcal{A}_{\Phi_f}$ and $\mathcal{A}_{\Phi_b}$ on their vertices respectively.

### B.2 Mesh Triangulation

In this section, we detail the meshing process of ISP. We first create a square 2D mesh $\mathbf{T}$ for $\Omega$ as shown on the left of Fig. 23. Given the latent code $\mathbf{z}$ of a specific garment, for each vertex $v \in V_\Omega$, we compute its signed distance value $s$ and edge label $c$ with $(s, c) = \mathcal{I}_\Theta(v, \mathbf{z})$. The 2D front and back panel meshes $\mathbf{T}_{P_f}$ and $\mathbf{T}_{P_b}$ are constructed by keeping vertices of $\mathbf{T}$ with negative signed distance (the blue region of the colored gird in Fig. 23) and those that have positive signed distance but belong to the edges crossing the 0 iso-level (the gray region of the colored gird in Fig. 23). For the later ones, we adjust their positions from $v$ to $\hat{v} = v - s(v, \mathbf{z})\nabla s(v, \mathbf{z})$ to project them to the zero level set (the blue line). Finally, we query $\mathcal{A}_{\Phi_f}$ and $\mathcal{A}_{\Phi_b}$ on each vertex of $\mathbf{T}_{P_f}$ and $\mathbf{T}_{P_b}$ respectively to lift them to 3D, giving us the front and back surfaces $\mathbf{T}_{S_f}$ and $\mathbf{T}_{S_b}$ as shown on the right of Fig. 23.

To sew the lifted front and back surfaces $\mathbf{T}_{S_f}$ and $\mathbf{T}_{S_b}$, we perform triangulation with the help of panel meshes $\mathbf{T}_{P_f}$ and $\mathbf{T}_{P_b}$. As illustrated in Fig. 24, we group the boundary vertices of $\mathbf{T}_{P_f}$ and

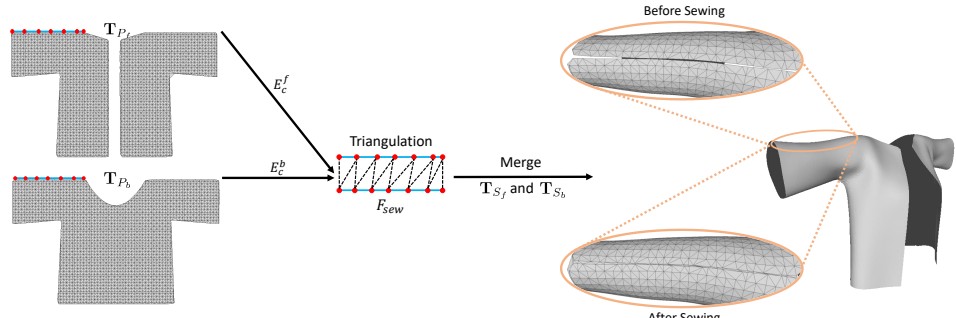

Figure 24: **Sewing process.** Left: the boundary vertices belonging to the sewing edges $E_c^f$ and $E_c^b$ are marked in red. Middle: The triangulation is performed between $E_c^f$ and $E_c^b$ to create new faces $F_{sew}$. Right: The mesh generated by merging the front and back surfaces $\mathbf{T}_{S_f}$ and $\mathbf{T}_{S_b}$ with $F_{sew}$.

$\mathbf{T}_{P_b}$ whose predicted labels are the same ($c$, with $c > 0$) to form the sewing edges $E_c^f$ and $E_c^b$ for the front and back panels separately. Then we create faces $F_{sew}$ between the vertices of $E_c^f$ and $E_c^b$, and use $F_{sew}$ to merge the meshes of $\mathbf{T}_{S_f}$ and $\mathbf{T}_{S_b}$, which gives us the final assembled garment mesh.

### B.3 Proof of the Differentiability of ISP

According to the *Theorem 1* of [50], for an SDF $s$ and the point $\mathbf{x}_0$ lying on the 0 iso-level $l = \{\mathbf{q}|s(\mathbf{q}, \mathbf{z}) = 0, \mathbf{q} \in \Omega\}$, we have

$$\frac{\partial \mathbf{x}_0}{\partial s} = -\nabla s(\mathbf{x}_0, \mathbf{z}). \tag{13}$$

For point $\mathbf{x}$ lying on the $\alpha$ iso-level, we can have $s(\mathbf{x}, \mathbf{z}) = \alpha$, where $\alpha$ is a constant. Let $s_\alpha = s - \alpha$, then $\mathbf{x}$ lies on the 0 iso-level of $s_\alpha$. Based on Eq. 13, we can have

$$\frac{\partial \mathbf{x}}{\partial s_\alpha} = -\nabla s_\alpha(\mathbf{x}, \mathbf{z}) = -\nabla s(\mathbf{x}, \mathbf{z}). \tag{14}$$

Assume $v$ is a vertex of the mesh $\mathbf{T}_G$ reconstructed by ISP and $\mathbf{x}$ is the point on the UV space that satisfies $v = \mathcal{A}_\Phi(\mathbf{x}, \mathbf{z})$, then it holds that

$$\frac{\partial v}{\partial \mathbf{z}} = \frac{\partial \mathcal{A}_\Phi}{\partial \mathbf{z}}(\mathbf{x}, \mathbf{z}) + \frac{\partial \mathcal{A}_\Phi}{\partial \mathbf{x}} \frac{\partial \mathbf{x}}{\partial \mathbf{z}}(\mathbf{x}, \mathbf{z}), \tag{15}$$

$$= \frac{\partial \mathcal{A}_\Phi}{\partial \mathbf{z}}(\mathbf{x}, \mathbf{z}) + \frac{\partial \mathcal{A}_\Phi}{\partial \mathbf{x}} \frac{\partial \mathbf{x}}{\partial s_\alpha} \frac{\partial s_\alpha}{\partial s} \frac{\partial s}{\partial \mathbf{z}}(\mathbf{x}, \mathbf{z}). \tag{16}$$

Since $\frac{\partial s_\alpha}{\partial s} = 1$, we can substitute Eq. 14 into Eq. 16 to derive that

$$\frac{\partial v}{\partial \mathbf{z}} = \frac{\partial \mathcal{A}_\Phi}{\partial \mathbf{z}}(\mathbf{x}, \mathbf{z}) - \frac{\partial \mathcal{A}_\Phi}{\partial \mathbf{x}} \nabla s(\mathbf{x}, \mathbf{z}) \frac{\partial s}{\partial \mathbf{z}}(\mathbf{x}, \mathbf{z}). \tag{17}$$

### B.4 Garment Draping

**Single Layer Draping.** In Fig. 25, we illustrate the pipeline for the single layer draping, which relies on the diffused LBS of SMPL [60] to get the initial rough estimate of the garment shape and the displacement map output by $\mathcal{D}_s$ to refine it.

**Multi-Layer Draping.** Our layering network $\mathcal{D}_m$ can be applied to multiple garments iteratively to resolve collisions between them. More specifically, consider $K$ garments $[G_1, G_2, ..., G_K]$ that are already draped individually by single layer draping network $\mathcal{D}_s$. Their subscripts denote their draping order, with smaller ones being closer to the body. We can obtain their rest state maps $[M_r^1, M_r^2, ..., M_r^K]$ as described in Sec. 3.2 of the main paper, and apply Algorithm 1 for layering them by iterating on garments following their draping order.

Note that we only train $\mathcal{D}_m$ on *one* pair of garments individually draped by $\mathcal{D}_s$. Therefore, it can only resolve the intersections happening at the same layer, which leads to an extra inner loop in Algorithm

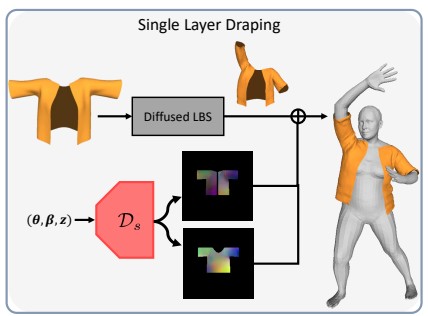

Single Layer Draping

Figure 25: **Single layer draping.** The rest-state garment is first deformed by the diffused LBS to get the initial shape, and then refined by the displacement maps predicted by $\mathcal{D}_s$.

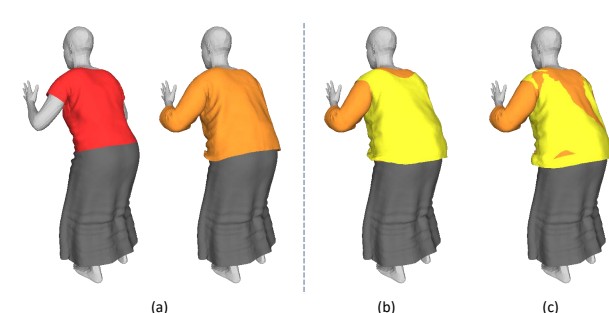

(a) (b) (c)

Figure 26: **Layering.** (a) We drape a red and an orange shirts on the body. A third one (yellow) is draped by (b) Algorithm 1 and (c) by naively applying $\mathcal{D}_m$ to it.

---

**Algorithm 1:** Multi-layer Draping

---

**Require :** Function `ForceMap(a, b)` that computes the force map for $a$ by taking $b$ as the underlying layer; Function `PositionMap(a)` that computes the 2D position map of $a$; Layering network $\mathcal{D}_m$.

**Input** : An ordered set of garments $[G_1, G_2, ..., G_K]$; Rest state maps for each garemnts $[M_r^1, M_r^2, ..., M_r^K]$.

**Output :** Layered garments $\{\tilde{G}_1, \tilde{G}_2, ..., \tilde{G}_K\}$ without intersections.

---

1 **for** $i \leftarrow 1$ **to** $K$ **do**
2   $\tilde{G}_i \leftarrow G_i$;
3   **for** $j \leftarrow i + 1$ **to** $K$ **do**
4    $M_f^j \leftarrow$ `ForceMap`$(G_j, \tilde{G}_i)$ ;       /* Eq. (9) in the main paper */
5    $M_d^j \leftarrow$ `PositionMap`$(G_j)$;
6    Update vertex positions of $G_j$ by $\mathcal{D}_m(M_r^j, M_d^j, M_f^j)$ ;

---

1 that moves all the subsequent garments to the same $j$-th layer. Otherwise, intersections cannot be completely resolved when applying $\mathcal{D}_m$ to two garments lying on the different layers as illustrated in Fig. 26(c).

## B.5 Recovering Multi-Layered Garments from Images

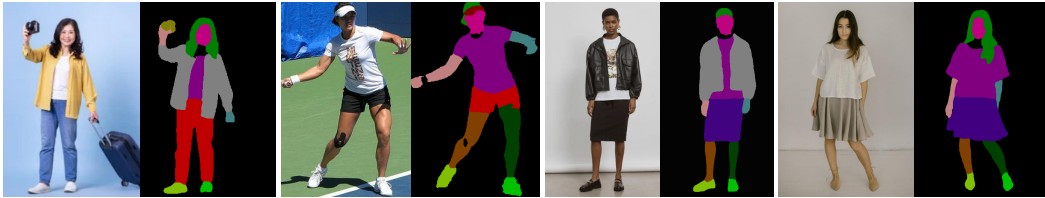

Figure 27: Segmentation masks obtained from [57]. Jackets are marked in gray, shirts in purple, skirts in dark purple, trousers in red, and body parts in other colors.

In Fig. 27, we show the segmentation masks used for the optimization in Eq. (11) in the main paper. The optimization is performed from the outer garment to the inner one, i.e., jacket (if detected) $\rightarrow$ shirt $\rightarrow$ trousers (or skirt). More specifically, for the first example shown in Fig. 27, we have the detected segmentation mask $\mathbf{S}_1$, $\mathbf{S}_2$ and $\mathbf{S}_3$ for the jacket, the shirt and the trousers respectively. We first initialize a latent code $\mathbf{z}_1$ for the jacket and perform the following optimization to recover its mesh

$$\mathbf{z}_1^* = \arg\min_{\mathbf{z}_1} L_{\text{IoU}}(R(\mathcal{G}(\mathbf{B}, \boldsymbol{\Theta}, \mathbf{z}_1) \oplus \mathcal{M}(\mathbf{B}, \boldsymbol{\Theta})), \mathbf{S}_1) , \tag{18}$$

$$\mathcal{G}(\mathbf{B}, \mathbf{\Theta}, \mathbf{z}_1) = \mathcal{D}(\mathbf{B}, \mathbf{\Theta}, \mathbf{z}_1, \mathbf{T}_G(\mathbf{z}_1)) \,, \tag{19}$$

Note that the rendered mask is obtained by setting the colors of the jacket and the body mesh vertices to white and black respectively. Then we fix $\mathbf{z}_1$ and initialize a new latent code $\mathbf{z}_2$ for the shirt and perform

$$\mathbf{z}_2^* = \arg\min_{\mathbf{z}_2} L_{\text{IoU}}(R(\mathcal{G}(\mathbf{B}, \mathbf{\Theta}, \mathbf{z}_{1:2}) \oplus \mathcal{M}(\mathbf{B}, \mathbf{\Theta})), \mathbf{S}_2) \,, \tag{20}$$

$$\mathcal{G}(\mathbf{B}, \mathbf{\Theta}, \mathbf{z}_{1:2}) = \mathcal{D}(\mathbf{B}, \mathbf{\Theta}, \mathbf{z}_1, \mathbf{T}_G(\mathbf{z}_1)) \oplus \mathcal{D}(\mathbf{B}, \mathbf{\Theta}, \mathbf{z}_2, \mathbf{T}_G(\mathbf{z}_2)), \tag{21}$$

to recover the mesh of the shirt. The optimization for the trousers is similar: fixing $\mathbf{z}_1$ and $\mathbf{z}_2$; initializing the latent code $\mathbf{z}_3$ for the trousers; performing

$$\mathbf{z}_3^* = \arg\min_{\mathbf{z}_3} L_{\text{IoU}}(R(\mathcal{G}(\mathbf{B}, \mathbf{\Theta}, \mathbf{z}_{1:3}) \oplus \mathcal{M}(\mathbf{B}, \mathbf{\Theta})), \mathbf{S}_3) \,, \tag{22}$$

$$\mathcal{G}(\mathbf{B}, \mathbf{\Theta}, \mathbf{z}_{1:3}) = \mathcal{D}(\mathbf{B}, \mathbf{\Theta}, \mathbf{z}_1, \mathbf{T}_G(\mathbf{z}_1)) \oplus \mathcal{D}(\mathbf{B}, \mathbf{\Theta}, \mathbf{z}_2, \mathbf{T}_G(\mathbf{z}_2)) \oplus \mathcal{D}(\mathbf{B}, \mathbf{\Theta}, \mathbf{z}_3, \mathbf{T}_G(\mathbf{z}_3)). \tag{23}$$

### B.6 Loss Terms, Network Architectures and Training

**Loss Terms.** The Chamfer distance loss $\mathcal{L}_{CHD}$ of Eq. (4) in the main paper is formulated as

$$\mathcal{L}_{CHD} = \sum_{\mathbf{x} \in P} \min_{\mathbf{X} \in \mathcal{S}} ||\mathcal{A}_\Phi(\mathbf{x}, \mathbf{z}) - \mathbf{X}||_2^2 + \sum_{\mathbf{X} \in \mathcal{S}} \min_{\mathbf{x} \in P} ||\mathcal{A}_\Phi(\mathbf{x}, \mathbf{z}), \mathbf{X}||_2^2, \tag{24}$$

where $P$ is the panel and $\mathcal{S}$ is the ground truth surface mesh.

The normal consistency loss $\mathcal{L}_{normal}$ of Eq. (4) in the main paper is formulated as

$$\mathcal{L}_{normal} = \sum_{\mathbf{x} \in Fc_P} (1 - n_f(\mathcal{A}_\Phi(\mathbf{x}, \mathbf{z})) \cdot \mathbf{n}_{\mathbf{X}^*}) + \sum_{\mathbf{X} \in Fc_\mathcal{S}} (1 - n_f(\mathcal{A}_\Phi(\mathbf{x}^*, \mathbf{z})) \cdot \mathbf{n}_{\mathbf{X}}), \tag{25}$$

$$\mathbf{X}^* = \operatorname*{argmin}_{\mathbf{X} \in Fc_\mathcal{S}} ||\mathbf{X} - \mathcal{A}_\Phi(\mathbf{x}, \mathbf{z})||_2, \quad \mathbf{x}^* = \operatorname*{argmin}_{\mathbf{x} \in Fc_P} ||\mathbf{X} - \mathcal{A}_\Phi(\mathbf{x}, \mathbf{z})||_2 \tag{26}$$

where $Fc_P$ and $Fc_\mathcal{S}$ are the face centers of the panel mesh and the surface mesh, and $\mathbf{n}_{\mathbf{X}}$ represents the normal of $\mathbf{X}$. $n_f(\cdot)$ is the function that computes the normal for the face that $\mathcal{A}_\Phi(\mathbf{x}, \mathbf{z})$ belongs to.

**Network Architectures.** For each garment category, i.e. shirts, skirts, and trousers, we train one separate set of networks $\{\mathcal{I}_\Theta, \mathcal{A}_\Phi, \mathcal{D}_s\}$. $\mathcal{D}_m$ is shared by all garment categories. Our models are implemented as the following.

- Pattern parameterization network $\mathcal{I}_\Theta$: We use two separate networks $\mathcal{I}_{\Theta_f}$ and $\mathcal{I}_{\Theta_b}$ to learn the pattern parameterization for the front and back panels. Each of them is implemented as an MLP with Softplus activations.

- UV parameterization network $\mathcal{A}_\Phi$: We use two separate networks $\mathcal{A}_{\Phi_f}$ and $\mathcal{A}_{\Phi_b}$ to learn the UV parameterization for the front and back surfaces. Both of them have the same architecture, which is a 7-layer MLP with a skip connection from the input layer to the middle and Softplus activations.

- Latent code $\mathbf{z}$: The dimension is 32.

- Diffuse skinning weight model $w$: A 9-layer MLP with leaky ReLU activations and an extra Softmax layer at the end to normalize the output.

- Displacement network $\mathcal{D}_s$: A 10-layer MLP with a skip connection from the input layer to the middle and leaky ReLU activations.

- Layering network $\mathcal{D}_m$: A U-Net with 4 convolution blocks and 4 deconvolution blocks.

**Training.** We use the Adam [**?** ] optimizer for training our networks. The batch sizes, the learning rates and the numbers of iterations for training are summarized in Table. 7. The hyperparameters of the training losses are summarized in Table. 8. $\mathcal{I}_\Theta$, $\mathcal{A}_\Phi$, $w$ and $\mathcal{D}_m$ are trained with a TESLA V100 GPU, while $\mathcal{D}_s$ is trained with 3 GPUs. During the training of $\mathcal{D}_m$, we randomly select two garments as the outer and inner layers, and let the model learn to resolve intersections between them.

| Network | $\mathcal{I}_\Theta$ | $\mathcal{A}_\Phi$ | $w$ | $\mathcal{D}_s$ | $\mathcal{D}_m$ |
|---|---|---|---|---|---|
| Learning Rate | $10^{-4}$ | $10^{-4}$ | $10^{-4}$ | $5 \times 10^{-5}$ | $10^{-4}$ |
| Batch Size | 50 | 50 | 6000 | 30 | 6 |
| Iterations | 70000 | 70000 | 2000 | 20000 | 30000 |

Table 7: Training hyperparameters.

| Loss | $\mathcal{L}_\mathcal{I}$ | | $\mathcal{L}_\mathcal{A}$ | | $\mathcal{L}_{\mathcal{D}_m}$ | | |
|---|---|---|---|---|---|---|---|
| | $\lambda_{CE}$ | $\lambda_{reg}$ | $\lambda_n$ | $\lambda_c$ | $\lambda_g$ | $\lambda_r$ | $\epsilon$ |
| Value | 0.01 | 0.001 | 0.01 | 1 | 0.5 | 1 | 0.005 |

Table 8: Training loss hyperparameters for $\mathcal{L}_\mathcal{I}$, $\mathcal{L}_\mathcal{A}$ and $\mathcal{L}_{\mathcal{D}_m}$.

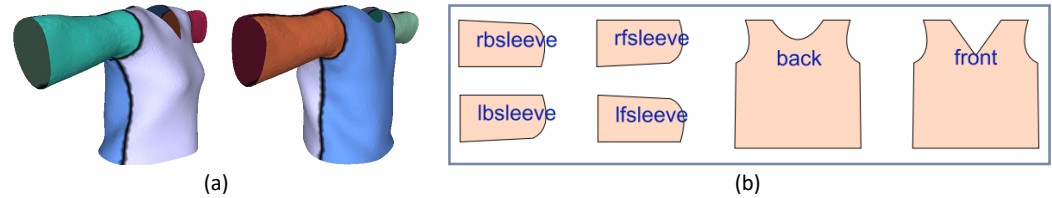

Figure 28: **The 6-panel sewing pattern**. (a) The 3D mesh surface for a shirt. The corresponding surface for each panel is denoted in different colors. (b) The six 2D panels for the front, the back, the right front/back sleeves and the left front/back sleeves.

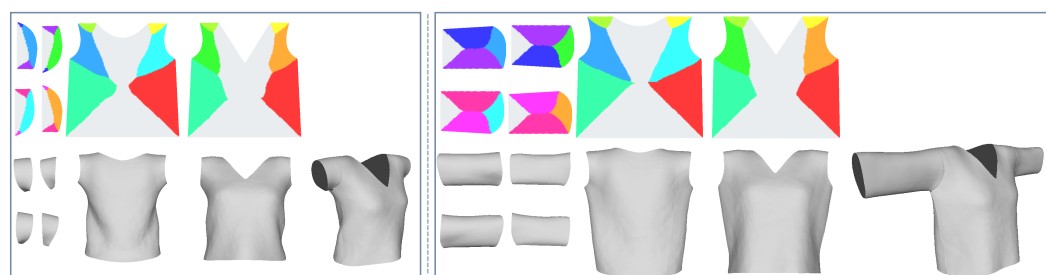

Figure 29: **Reconstruction with 6 panels**. For both the left and right examples, we show the reconstructed panels on the top and the reconstructed surfaces and the sewed meshes at the bottom. Colors on panels denote edge labels predicted by $\mathcal{I}_\Theta$.

## C   Extension to Sewing Patterns with More Panels

In our experiments, each garment's sewing pattern consists of two panels, the front and the back. However, our ISP can be extended to patterns with any number of panels. For example, we can train $\mathcal{I}_\Theta$ and $\mathcal{A}_\Phi$ on a database of sewing patterns with six panels as shown in Figure 28, using the same training protocol described in the main paper. After training, we can use them to reconstruct the panels and surfaces and produce the sewed mesh as illustrated in Fig. 29. Adding more panels does not result in better reconstructions. For this reason and for the sake of simplicity, we use a model with 2 panels as our default setting.

## D   Failure Cases

Fig. 30 presents draping results as the number of shirts increases. We observe that the model produces unrealistic deformation when the number of shirts is greater than four. This behavior occurs because our multi-layer draping model $\mathcal{D}_m$ is only trained on garments obtained by single layer draping as described in Section 3.2 of the main paper. In this scenario, the garments are relatively close to

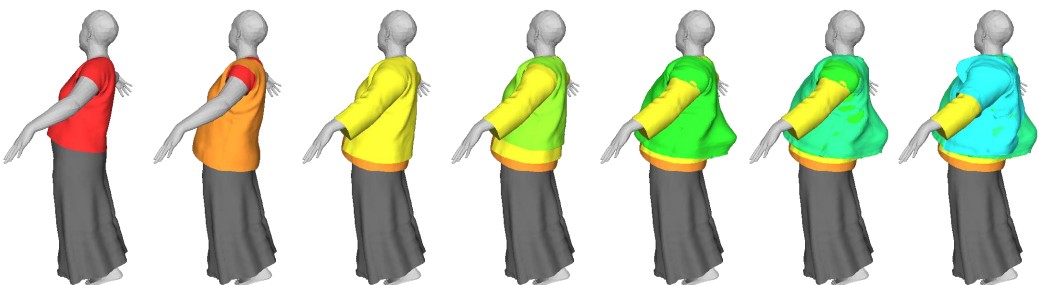

Figure 30: Draping increasingly many shirts: from 1 (left) to 7 (right).

the body. When applied to cases with more shirts (typically over four), the model may generate unpredictable results with the shirts moving far away from the body. However, we note that this issue can be resolved by finetuning the model progressively on layered garments. We also consider that wearing more than four shirts is not a common scenario.

