# OpenReview forum: "ISP: Multi-Layered Garment Draping with Implicit Sewing Patterns"
_NeurIPS.cc/2023/Conference — NeurIPS 2023 poster_

### Official Review · Reviewer_LvBx · 2023-06-19

**Soundness:** 2 fair
**Presentation:** 2 fair
**Contribution:** 2 fair
**Rating:** 3
**Confidence:** 4

**Summary:**

This paper proposes to drape multi-layered garments on SMPL-based human bodies with different poses and shapes. The method is inspired by the commonly adopted sewing pattern and learns to map garments from 2D panels, i.e. front and back panels, to 3D surfaces. The author first deforms the single-layered garments in rest pose according to SMPL skinning function, followed by the corrections of interpenetration between layers of garments. Experiments show the effectiveness and efficiency of the method. The author further shows some possible applications such as garment reconstructions from images and garment editing.

**Strengths:**

* An efficient method mapping implicit 2D sewing patterns to 3D garments with good performance.
* The qualitative results of draping multi-layered garments show less interpenetration.
* The method is differentiable and is able to be applied to inverse problems.

**Weaknesses:**

1. While this paper focuses on multi-layered garments, the experiments lack some quantitative results to support the effectiveness. For example, what is the rate of interpenetration between human body and multi-layered garments? What about the interpenetration between different layers of garments? Will different types of garments, such as dress or shirt, lead to different collision rate? How will the collision rates mentioned above change when the number of layers increase? In the meanwhile, please also provide more detail information about the test set for the above evaluation, such as the number of vertices or the resolution of the mesh.
2. As for the efficiency, the author provides comparisons during training process in Table 1. Could you provide similar comparisons about reconstruction time or model forward time during test?
3. The distribution of the human parameters in the dataset is unknown. How many different human bodies are sampled? What is the distribution of different gender in both training and test set?
4. Is there any overlapping between training set and test set? How many unseen garments and human bodies are included in test set?
5. While the previous method mentioned at L62 is only able to drape multi-layered garments in T-pose, could the author provide some comparisons also in T-pose? Since the method in this paper supports different poses of human and previous method at L62 did very similar work, the comparisons in T-pose are needed to support the effectiveness of the method.
6. The writings can be further improved. Some descriptions and captions are not clear.
7. Please change the tone of the sentence at L3. As discussed in related work at L62, previous method is able to drape multi-layered garments.
8. The layering network at L191-221 seems like a post-processing step, making the multi-layered draping more like a simple extension of single-layered draping.

**Questions:**

* For each garment, should we learn a unique latent vector z? Do we need to train different models for different garments?

**Limitations:**

While multi-layered settings are more challenging, this paper need more experiments and further explanations to support the contributions in terms of multi-layered garments.

First, though the method shows better qualitative results, some more experiments are needed to support the strengths of the method in terms of multi-layered settings, the efficiency, and even the robustness.
More comparisons between the existing methods mentioned at L62 are needed to prove the effectiveness of this method.

Second, as for the dataset, the number of samples seems not enough. The main concern is that whether the insufficient data would lead to some bias or overfitting and thus achieve better qualitative results. More quantitative results with clear explanations are needed.

Finally, the idea of getting interpenetration-free garments seems more like a post-processing step, where another model is trained and specifically aims to move the penetrated vertices out of the inner mesh. I am concerned regarding this idea. Since this makes the settings of multi-layered garments as a simple extension of single-layered garments, weakening the contributions in the field of multi-layered garments.

---

> ### Author Rebuttal · Authors · 2023-08-07
>
> Thank you for your valuable reviews. Below are our responses.
>
> 1. *The experiments of layering.*
>
> We conducted an experiment to further evaluate our layering model by measuring the intersection ratio between garment layers and between garments and the body. We generated 673 unseen bodies with unseen poses from [1] and shape parameters uniformly sampled from $[-2, 2]^{10}$. Each body was paired with 5 randomly generated unseen garments (1 skirt or 1 trousers with 4 shirts), where the 1st layer consisted of the skirt or trousers.
>
> The results, presented in Table 1 of the attached PDF file, display the intersection ratios. Diagonal values are the intersection ratio between the body and the $i$-th ($i=1,2,3,4,5$) layer, while other values are intersections between different garment layers. As prior works do not support layering, a direct comparison is impractical. The numbers inside and outside the bracket denote the results obtained with and without the layering procedure $\mathcal{D}_m$ respectively. The significantly lower intersection ratios with layering demonstrate the effectiveness of our model in handling intersections with the body and garments, even after layering five garments (~2%). The increase in intersections as more garments are layered is explained in L150-155 of the supplementary material, where we clarify the training of $\mathcal{D}_m$ is on single-layer draped garments, which are closer to the body. The average vertex numbers for shirts/skirt/trousers are 9.2k/9.4k/10.6k respectively.
>
> 2. *Similar comparisons about reconstruction time or model forward time during test.*
>
> As stated in L247, the time reported in the left of Table 1 in the main paper refers to the reconstruction time (or model forward time) for the garments from the training set, rather than the time taken for the training process. Furthermore, the labels 'Train'/'Test' in Table 1 indicate that the results are evaluated on the training/test set respectively, and not refer to the training/testing process. As shown in Table 2 of the attached PDF file, the reconstruction time during testing is comparable to those in the left of Table 1 of the main paper.
>
> 3. *The distribution of the human parameters.*
>
> We use SMPL to model the human bod. Its usual parameters $\mathbf{\Theta}$ and $\mathbf{B}$ control the body pose and shape respectively.
>
> Our training and evaluation protocol follows prior art [1,2], using the AMASS dataset. With 6519 poses for training and 673 unseen poses for testing, the dataset covers a wide range of human motions, including walking, running, jumping, arm/torso movements and dancing. Importantly, our self-supervised draping method, driven by a physics-based loss, eliminates the need of simulated or scanned garment data for training. This allows our model to easily generalize and extend to more pose data. During training and testing, we uniformly sample $\mathbf{B}$ from $[-2, 2]^{10}$.
>
> SMPL offers 3 body models: female, male and neutral. While our study focuses on the female model, as do many prior works [2-5], there is nothing specific about it and our approach applies just as well to the other two.
>
> 4. *Is there any overlapping between training set and test set?*
>
> No, there is no overlapping between training set and test set. As stated in L226-227 of the main paper, for the experiments of garment reconstruction, the test set comprises 20 shirts, 20 skirts, and 20 pairs of trousers that are not part of the training set (unseen). For garment draping, the test set contains 673 unseen $\mathbf{\Theta}$, along with $\mathbf{B}$ sampled from $[-2, 2]^{10}$ .
>
> 5. *Comparisons to [5].*
>
> There unfortunately is not easy way to generalize to [5] because the code is unavailable. In addition, it would only work in T-pose, whereas we generalize to different ones.
>
> 6. \& 7. *The writings and the tone of L3.*
>
> We will revise our paper to make it clearer, and rephrase L3 to '*However, they are either unable to handle multi-layered clothing, which is prevalent in everyday dress, or restricted to bodies in T-pose*'.
>
> 8. *The multi-layered draping like an extension of single-layered draping.*
>
> The layering network is indeed a straightforward extension of single-layered draping precisely because our single-layered model is designed to make this extension easy, which we view as part of our contribution. Single-layer draping provides a good starting point for our layering model. By predicting corrective displacement for vertices, rather than directly regressing vertex positions, we can ease the training process. This is important because our model is *self-supervised* using a physics-based loss, and directly regressing vertex positions would cause the training to collapse and not converge.
>
> 9. *The latent vector and model for different garments.*
>
> We use a single network to handle a wide range of garments of varying topology and geometry. Each garment is associated to a unique latent vector $\textbf{z}$ learned by auto-decoding, as in [6]. During training, the latent vectors for garments in the database are randomly initialized and optimized alongside the weights of our ISP network $\mathcal{I}_{\Theta}$ using backprop. For unseen garments, we randomly initialize a new $\textbf{z}$, and optimize it while keeping the network frozen by fitting it to observations like image segmentations or 3D meshes.
>
> For our draping models $\mathcal{D}_s$ and $\mathcal{D}_m$, we learn a single generic network separately that can handle various garments rather than multiple garment-specific networks. This makes our approach more scalable, without having to train and maintain a separate network for each garment.
>
> ### References
> *[1] N. Mahmood, et al. AMASS. ICCV 2019.*
> *[2] I. Santesteban, et al. SNUG. CVPR 2022.*
> *[3] L. Luigi, et al. DrapeNet. CVPR 2023.*
> *[4] I. Santesteban, et al. VTO Garment Collisions. CVPR 2021.*
> *[5] I. Santesteban, et al. Ulnef. NIPS 2022.*
> *[6] J. Park, et al. Deepsdf. CVPR 2019.*

---

> > ### Comment · Reviewer_LvBx · 2023-08-16
> > **Reply to rebuttal**
> >
> > Thank you for your explanations.
> > However, I still find the technical novelty is limited.
> > While the author claims that "multi-layered draping" is one of the main contribution, two key questions are unable to solve and cannot well support the contributions to multi-layered settings.
> >
> > First, since ULNeF [1] is the most related work, which also tries to solve multi-layered garments draping with state-of-the-art performance in 2022, the author should provide the fair comparisons with ULNeF. However, no comparison between ULNeF and the proposed method is provided, such as in easier settings T-pose. Thus, the effectiveness of the proposed method cannot be proved. And there is no evidence to prove that the proposed method can achieve better performance than existing work in terms of multi-layer draping.
> >
> > Second, as the author replied in Q8, the module designed for multi-layered settings is straightforward extension of the work, while this idea, which tries to predict displacement to reduce the interpenetration, is also widely applied in previous work such as the GarSim [2]. Other work, such as the [3], only regard the multi-layered settings as simple extension instead of the main contribution, with much easier way to extend to multi-layered garment animation. The straightforward extension is more like a post-processing step, which can also be achieved even with numerical methods in TailorNet[4] and obtains interpenetration-free performance.
> >
> > In short, the paper is not ready due to the limited technical novelty. The straightforward extension (post-processing step) is not enough to be regarded as a main contribution.
> >
> > [1] Santesteban, et al. ULNeF, NeurIPS2022
> > [2] Tiwari, et al. GarSim, WACV2023
> > [3] Meng, et al. MotionGuided, ToG2022
> > [4] Chaitanya, et al. TailorNet, CVPR2020

---

> > > ### Author Response · Authors · 2023-08-16
> > >
> > > Thank you for your feedbacks.
> > >
> > > First, we would like to point out that we have addressed all the reviewer’s comments, save the one about the comparison with ULNeF.
> > >
> > > Second, we have to disagree with the reviewer’s assessment of ULNeF: The authors provide neither the code nor numbers on standard benchmarks. This makes a comparison almost impossible and we view this as something that is lacking from the ULNeF paper. Furthermore, it only comprises results on a T-Pose, without any evidence that the proposed approach would work under more challenging circumstances. In fact, the ULNeF authors themselves mention the difficulty in extending their formulation to more complex poses is one of their limitations. By contrast, this limitation does not apply to us and we show results in a much broader range of situations.
> > >
> > > Regarding the layering, we propose a novel representation that enables us to simplify the complex problem of layering, and treat it as a straightforward extension of the single-garment version of our approach. This is valuable contribution because it provides a unified framework for handling multi-layered draping by converting the three-dimensional threading problem to the two-dimensional template level for processing, which does not appear in prior art. Furthermore, our proposed representation offers additional advantages beyond multi-layered draping. It is differentiable, and enables latent space interpolation and fitting observations, which expands the applicability of our approach to garment generation, editing, and recovery tasks.
> > >
> > > Finally, the method of [3] involves learning a garment-specific model using ground truth simulation data. It can be extended to cases with a limited number of garments---two in the paper---as long as ground truth data is available. However, handling a large collection of garments would require collecting simulation data for different garment combinations and train separate models for each case, which would be prohibitively costly and impractical. In contrast, our method leverages a single generic model and self-supervised learning that can handle a wide range of garments without the need for additional data collection or separate model training, which is another valuable contribution.

---

> > > > ### Comment · Reviewer_LvBx · 2023-08-18
> > > > **Reply to Response**
> > > >
> > > > Thank you for your reply.
> > > >
> > > > Most of the weaknesses are solved except the two key problems/limitations, which I have mentioned in the first review and summarise in the following based on the rebuttal:
> > > > 1. Technical novelty about the multi-layered settings, which is claimed as the main contribution of this paper, is limited.
> > > >     1. First, the author claims that the module dealing with multi-layered clothes is a straight forward extension. The paper thus can be regarded as “single-layered draping + extension”, which is composed of two separate module and is not fully end-to-end.
> > > >     2. Second, the simple extension, which is a post-processing step, can also be replaced by numerical methods, which are adopted in some existing work, such as TailorNet [1] and Layered-Garment Net [2], where the latter uses the hard-coded post-processing step to correct the collisions and generate the interpenetration-free multi-layered garment dataset. The numerical methods in those work seem stable and flexible, without the need of training any model. From the visual results, the numerical methods also can obtain smooth and collision-free garments. There is no evidence showing that the proposed extension is better than the numerical methods. Based on the above observation, the importance of the proposed extension is thus less significant.
> > > >     3. Third, if simply adopting the post-processing step is enough to deal with multi-layered garments, the multi-layered settings seem less challenging and are only simple extensions of single-layered garments.
> > > > 2. Important comparison with existing work, I.e. ULNeF [3] in 2022, is missing. There is no evidence to prove that the proposed method can generate multi-layered garments of higher quality, such as the interpenetration rates, even in T-pose. And since ULNeF is the most related work, the comparison is also necessary.
> > > >
> > > > I am still not convinced about the novelty of this paper, which is the most important point. The simple extension, which is also part of existing work [4], has limited contribution. As for the comparisons with ULNeF, though the author lists some reasons to try to avoid the comparisons, the implementation of ULNeF is necessary due to the importance and relevance to this paper.
> > > >
> > > > Some more comments about the application of the multi-layered garments. Layered-Garment Net [2] reconstructs the multi-layered cloths from single images. SMPLicit [5] proposes a fully differentiable generative model for garment editing, and is able to handle multi-layered garments. While all tasks mentioned in paper have been explored by many existing methods, only "garment reconstruction" part, which is not able to support the contributions of this paper, has comparisons with previous work.
> > > >
> > > > [1] Chaitanya, et al. TailorNet, CVPR2020 [2] Aggarwal, et al. Layered-Garment Net, ACCV2022. [3] Santesteban, et al. ULNeF, NeurIPS2022. [4] Tiwari, et al. GarSim, WACV2023. [5] Corona, et al. SMPLicit, CVPR2021.

---

> > > > > ### Author Response · Authors · 2023-08-19
> > > > >
> > > > > The reviewer keeps on insisting that we should compare to ULNeF, even though we have explained several times that doing this comparison would be very hard. The reviewer suggests that we should re-implement this method, whereas we believe the onus of providing the code rests with the authors.
> > > > >
> > > > > Even more crucially, we are not even convinced a formal comparison is necessary. Our formulation allows us to handle arbitrary poses.  In contrast at the bottom of page 8 in the “limitations” paragraph, the ULNeF authors write "The proposed approach for VTO using ULNeFs has only been validated with garments in T-pose. The root of this limitation is the difficulty in extending the formulation based on covariant fields to more complex poses”.  In other words, our formulation eliminates this problem and makes the draping of multiple garments in arbitrary pose straightforward. Instead of using covariant fields as in ULNeF, we represent garments by the combination of 2D SDF and a 2D-to-3D mapping fuction, and resolve garment interpenetrations by the generic network trained with physics-based loss in a self-supervised fashion. We do not understand why the reviewer refuses to accept this as a contribution.
> > > > >
> > > > > Neither do we understand the reviewer's assessment of our multi-layer module. Our pipeline is *fully end-to-end differentiable*, from the ISP representation module, to the single-layer module, and the multi-layered module.
> > > > >
> > > > > With regards to the other papers mentioned in the reviewer’s latest comment:
> > > > >
> > > > > - TailorNet is designed to address single-layer draping. There is no evidence to suggest its suitability for multi-layered draping. As a template-based method, its representation ability is limited. Its numerical approach is effective in resolving small collisions between the garment and the underlying mesh. However, as indicated in the attached PDF file's Table 1 (bottom), the interpenetration ratio between garments draped by the single-layer model can be substantial (over 20\%). In such cases, simply moving the outer layer's vertices outward does not yield physically realistic results. In contrast, our model is trained to minimize physics-based loss applied in cloth simulation [Narain2012], resulting in more physically plausible outcomes.
> > > > >
> > > > > - Layered-Garment Net (LGN) focuses on garment recovery from images, instead of garment draping. It resolves interpenetration through finetuning the weights of the SDF network, which involves multiple iterations and consequently becomes time-consuming. In LGN, how to drape a set of garments over an arbitrary body with it is not explicitly addressed. However, thanks to our novel representation and its differentiability, our method can be applied effectively to both garment recovery and draping tasks. By leveraging our multi-layered draping module, we can resolve interpenetration by a single forward pass, which is more efficient than the finetuning step of LGN.
> > > > >
> > > > > - SMPLicit represents garments using watertight meshes reconstructed from SDF. However, the garment mesh should be an open surface, which aligns with our methodology. Furthermore, SMPLicit is not able to handle multi-layered garments, as its reconstructed watertight meshes collide with each other. These two issues limit its potential for downstream applications such as cloth simulation.
> > > > >
> > > > >
> > > > > *Rahul Narain, Armin Samii, and James F O’brien. Adaptive Anisotropic Remeshing for Cloth Simulation. ACM Transactions on Graphics (Proc. SIGGRAPH Asia), 31(6):1–10, 2012.

---

### Official Review · Reviewer_5F53 · 2023-07-06

**Soundness:** 3 good
**Presentation:** 3 good
**Contribution:** 3 good
**Rating:** 6
**Confidence:** 4

**Summary:**

The authors address the task of draping individual multi-layer garments on human body models. In this context, they introduce a respective garment representation suitable for this task. Garments are represented as a set of individual 2D panels whose shape is defined based on a signed distance function (in more detail, the zero-crossing of a function with 2D location and latent vector specific to each garment as inputs). For each 2D panel, a 2D-to-3D mapping (conditioned on the 2D location and latent vector) is used to map the 2D panel to the 3D garment surface, while enforcing continuity across panels. In addition, draping networks are trained to allow draping multiple garments on human bodies (represented based on SMPL model) in different poses.

The authors provide quantitative and qualitative comparisons that indicate some potential of the proposed approach.


**Strengths:**

Technical soundness:
- The approach seems reasonable and the results indicate some improvement over the competing techniques.

Evaluation:
- The authors provide quantitative and qualitative evaluations with comparisons to alternatives.
- The supplemental provides further experiments as well as limitations.

Exposition:
- The paper is well-structured and good to follow. Figures/tables and respective captions are informative.
- Text quality is good.
- The authors provide a comprehensive supplemental.

References:
- References seem ok, but I am not an expert in this field.



**Weaknesses:**

Technical soundness:
- Table 1 only shows results for shirts. What are the results for the other categories like?
- The variations in the used dataset are not discussed in detail. More details on this would be interesting and allow a better assessment on the potential and remaining challenges of the presented approach.

Evaluation:
- The evaluation could have been improved by also showing examples or highlighting what works particularly well in comparison to other approaches and what still remains challenging for the presented method.
- Providing a visualization of the spatial distribution of the Chamfer distance across the surface would show where the errors are larger and where the errors are small. This could help to show the accuracy a bit better, also when comparing to others.
- The user study is quite simple regarding the fact that only direct preferences have been checked for. More detailed subquestions could have provided insights on what exactly the participants in the study did not yet like and where further improvement is needed.
- For Figure 6, zoom-ins could be used to highlight improvements/differences between methods.

Reproducibility:
- The paper presents a relatively complex system. This might complicate reproducibility. It is not clear whether code and data will be made available.


**Questions:**

Please address my comments under ‘Weaknesses’.

**Limitations:**

Limitations and failure cases have only been discussed in the supplemental.

---

> ### Author Rebuttal · Authors · 2023-08-07
>
> We thank you for your valuable reviews. Below are our responses to your comments.
>
> 1. *Table 1 only shows results for shirts. What are the results for the other categories like?*
>
> The results for trousers and skirts can be found in Table 1 and Table 2 of the supplementary material respectively, where our method shows similar performance with higher accuracy and faster inference speed than UDF.
>
> 2. *The variations in the used dataset are not discussed in detail.*
>
> We apologize for the oversight regarding the detailed discussion about the dataset in our paper.
>
> For the generation of sewing patterns and corresponding 3D garment meshes, we used the software developed by [Korosteleva2021]. Our dataset encompasses three garment categories: shirts, skirts, and trousers. In [Korosteleva2021], shirts are parameterized by \{length, width, hem width, collar width, front/back collar depth, sleeve connection width, sleeve opening width, sleeve length\}, while skirts and trousers are parameterized by \{length, width, front/back curve\} and \{length, crotch depth, hem width\}, respectively. The values of these parameters were uniformly sampled from pre-defined ranges.
>
> To create our training set, we generated 400 shirts, 300 skirts, and 200 pairs of trousers. Additionally, we prepared a separate testing set, consisting of 20 shirts, 20 skirts, and 20 pairs of trousers. In the supplementary material, Figures 2, 3, and 4 showcase the variations in garment shape and style that were incorporated into the dataset.
>
> In line with the methodology described in [Santesteban2022], we relied on the AMASS dataset [Mahmood2019] for training and evaluating our draping models. The AMASS dataset offers a diverse range of poses, containing activities such as walking, running, jumping, arm movements, torso movements, dancing, and more, which ensures comprehensive coverage of various human motions.
>
> 3. *The evaluation could have been improved by also showing examples or highlighting what works particularly well in comparison to other approaches and what still remains challenging for the presented method.*
>
> We conducted comparisons between our method and prior approaches in garment reconstruction, draping, and recovery. Our results demonstrate that our method is more accurate and faster than UDF in the context of reconstruction. Additionally, our method is able to handle multi-layered draping, which is not achieved by any other existing works. A more detailed comparison and comprehensive analysis can be found in the supplementary material (Sec. 1 and 4), where we provide further insights, limitations, and discussion of failure cases.
>
> 4. *Providing a visualization of the spatial distribution of the Chamfer distance.*
>
> The visualization of the spatial error distribution can be found in Figure 1 of the attached PDF file. In this figure, we compare the error distribution between our reconstructions and those produced by UDF. Our reconstructions exhibit lower error across the entire surface compared to UDF's. We will include this figure in the revised version of our paper.
>
> 5. *The user study is quite simple.*
>
> True but the simple interface and instructions we used allowed us to query many users from diverse backgrounds at a reasonably low cost. In future research, we will work on collecting finer feedback, while keeping a low user friction.
>
> 6. *For Figure 6, zoom-ins could be used to highlight improvements/differences between methods.*
>
> We will revise Fig. 6 for better visualization of the improvements.
>
> 7. *Reproducibility.*
>
> As mentioned in Line 44, we will release our codes and trained models.
>
> 8. *Limitations and failure cases have only been discussed in the supplemental.*
>
> We will revise our paper to incorporate the discussion of limitations and failure cases, as shown in Sec. 4 and Fig. 21 of the supplementary material, in the main paper.
>
>
> ### References
> *M. Korosteleva and S. Lee. Generating Datasets of 3D Garments with Sewing Patterns. In Advances in Neural Information Processing Systems, 2021.*
> *I. Santesteban, M.A. Otaduy, and D. Casas. SNUG: Self-Supervised Neural Dynamic Garments. In Conference on Computer Vision and Pattern Recognition, 2022.*
> *N. Mahmood, N. Ghorbani, N. F. Troje, G. Pons-Moll, and M. J. Black. AMASS: Archive of Motion Capture as Surface Shapes. In International Conference on Computer Vision, pages 5442–5451, 2019.*

---

> > ### Comment · Reviewer_5F53 · 2023-08-16
> > **Reply to rebuttal**
> >
> > I thank the authors for providing respective explanations to clarify several of the raised aspects.
> >
> > Regarding the answer to aspect 3, I want to stress that I saw the comparisons in the paper and supplemental for the review, but the way of visually highlighting the potential benefits for the respective examples is not as good and I would also expect the respective demonstration in the scope of more examples.
> >
> > Furthermore, when visualizing spatial deviations from the reference via the Hausdorff error, the authors should use a wider range in the color spectrum to better visualize deviations and depict respective results for more different examples and also for the results obtained with competing methods.
> >
> > I still wonder whether the provided information on the user study allow sufficient insights.
> >
> > Finally, most of all, I would be interested in the authors' comment regarding Reviewer LvBx's feedback on the technical novelty and the comparison to other multi-layer draping methods.

---

> > > ### Author Response · Authors · 2023-08-19
> > >
> > > Thanks for your feedback.
> > >
> > > 1. For the figures depicting spatial deviation, we will change the color code and select more representative examples for better visualizations in the final version of our paper.
> > >
> > > 2. With regards to previous art, reviewer LvBx’s major concern seems to be with the ULNeF method. In our response, we point out that
> > >     - The fact that the authors provide neither code nor quantitative results on accepted benchmarks makes comparison extremely difficult. We would have to re-implement their full method.
> > >     - Even more importantly, the paper only show results on T-Poses and acknowledges that their method may not be suitable to other poses. In the limitation section states “The proposed approach for VTO using ULNeFs has only been validated with garments in T-pose. The root of this limitation is the difficulty in extending the formulation based on covariant fields to more complex poses”. Our approach does not suffer any such limitation.
> > >
> > > 3. Reviewer LvBx discuses additional approaches in his latest comment. We responded to that as well.
> > >
> > > 4. In response to another comment, our method is end-to-end differentiable and trainable for both the single- and multiple- layer cases.

---

### Official Review · Reviewer_cVr6 · 2023-07-06

**Soundness:** 4 excellent
**Presentation:** 4 excellent
**Contribution:** 4 excellent
**Rating:** 7
**Confidence:** 4

**Summary:**

This paper introduces ISP, a novel system that can, for the first time, drape a 3D human body with multi-layer garments, without the need of physics-based simulation. Several technical contributions addresses the key aspects of garment draping. To enable learning of garment sewing patterns, the authors cleverly formulate the 2D patterns with 2D signed-distance fileds (SDFs) and label fields. An AtlasNet-style network maps the 2D sewing pattern to 3D, thereby draping clothing to the body. To achieve multi-layer draping, a intuitive mechanism derives a virtual repulsive force based on the 3D locations of draped garments, thereby resolving the collision of cloth.
Extensive experiments are performed to characterize the model in garment reconstruction, draping, image-based recovery and editing. In all the tasks, ISP has a clear advantage over existing methods, showing its great potential for many downstream tasks in visual computing.

**Strengths:**

I appreciate this paper for its strengths in many aspects:

- Significance of the problem. The paper addressed an important yet long-overlooked problem: automatic modeling, editing and perception of multi-layer clothing. This is a technically challenging problem. Being the first method tackling this (to my knowledge), this paper can potentially have a good impact in various relevant fields in both research and industry (fashion, movie-making etc.).

- Novelty. The technical contributions of this work are strong. It is clever design to leverage neural fields as a formulation for the seemingly discrete 2D sewing patterns, and to use AtlasNet for the draping problem -- both being creative ways to use recent powerful tools. The layer-wise draping is another great example of combining the power of CNNs with physics-inspired losses. In addition, the method provides a unified pipeline for garments of different topology, such that different garment pieces can be modeled in a standardized way -- another favorable property for machine-learning based approaches.

- Evaluation soundness. The experiments holistically validate the characteristics of the proposed model across multiple relevant tasks in vision and graphics, showing the versatility of the method: 3D garment reconstruction, garment draping, image-based garment reconstruction, and shape/texture editing. The implimentation details, runtime benchmarking are well documented. The user study of the draping quality makes the study more compelling.

- Model performance. To my knowledge, this is the first learning-based work that can drape multi-layer clothing on the bodies and handles cloth interpenetration adequately. Even in the single-layered setting, he proposed method has advantages over prior work in terms of less cloth-body intersecting, draping realism, reconstruction accuracy, and inference speed, all being important aspects of garment draping.

- Presentation. The paper is well-written, easy to follow, and technically sound. The math formulations are specific and clear. The illustrations are clear and well complements the text elaboration.

Overall, this is a very strong submission from which I've learned much, and I believe it can be highly impactful.

**Weaknesses:**

I do not spot major weaknesses in this paper but one observation. In the image-based garment recovery experiment (Sec. 4.4), although recovered garments are correct in their topology and coarse dimensions (e.g. length), but they do not match the details (overall deformation and wrinkles) as that in the image. This inherits the limitation from previous methods such as DrapeNet. Probably in future work a combination with photometric losses can result in more accurate wrinkle estimation?

**Questions:**

- Out of curiosity, how would the model perform when applied to garments that are too complex to be cut into simple front and back pieces (such as a robe or a double-breasted suits)?

- Currently the garment shape vector z is a single, global one for each garment. Would it increase the representation power (hence helpful  to reconstruction tasks) if the z-vector becomes pixel-aligned with the uv map, as done in e.g. NeuralActor (Liu et al., SIGGRAPH Asia 2021) and PoP (Ma et al., ICCV 2021)?

- Following the arguments in L.135, it is only necessary to deform the areas within the front and back panels instead of the full square of the UV map. However in single layer draping (L.184), the displacement MLP outputs the NxNx3 arrays, i.e. covering the entire square, which seems a bit inconsistent to me. To my understanding, in this case the values for out-of-panel areas are null or zero, is that right? If so, how to ensure that?

- Perhaps adding the dimensionality of z to Fig 2 to make it consistent with x?

**Limitations:**

The technical limitations are discussed in the conclusions and the authors provide an outlook to the solutions.

---

> ### Author Rebuttal · Authors · 2023-08-07
>
> Thank you for your appreciation of our work. We address your questions and comments as follows:
>
> 1. *A combination with photometric losses can result in more accurate wrinkle estimation.*
>
> True, incorporating such a loss could significantly enhance our current method. However, formulating an effective photometric loss is not easy:  The appearance of a garment in a single image is influenced not only by its geometry but also by factors such as texture and shading. The is something we plan to do  in future work.
>
> 2. *How would the model perform when applied to garments that are too complex to be cut into simple front and back pieces?*
>
> As demonstrated in Section 3 of the supplementary material, our method naturally extends to sewing patterns with multiple panels. This allows us to effectively handle the modeling of garments that are more complex than those represented by just two panels, i.e. the front and back panels. Thus, by incorporating additional panels, our method can accommodate intricate garment designs and accurately capture their details. However, it is worth noting that most common garments can easily be modeled by only two panels, and our model excels at handling them.
>
> 3. *Pixel-aligned latent codes would increase the representation power.*
>
> We agree that employing a pixel-aligned encoding strategy would enhance the representation power of our method. By adopting such a strategy, we would be able to preserve more intricate local details of the garment than just using a single global latent code. On the downside, however, a model using pixel aligned latent codes would a) require training images, that is, for each garment geometry we would need a realistic rendering, and b) be more suited for 3D reconstruction from images and less for tasks such as fitting 3D scans.
>
> 4. *In single layer draping (L.184), the displacement MLP outputs the NxNx3 arrays, i.e. covering the entire square, which seems a bit inconsistent to me. To my understanding, in this case the values for out-of-panel areas are null or zero, is that right? If so, how to ensure that?*
>
> Indeed, in our single layer draping model, the output is a $N\times N \times 3$ array representing the displacement $D_s$ that covers the entire square. To ensure that the out-of-panel areas, denoted as $x$, have zero values, we employ the signed distance function (SDF) of the corresponding garment. Specifically, for areas where the signed distance values are positive (i.e., $s(x,\textbf{z})>0$), we set $D_s(x)=0$. This corresponds to masking the output of the displacement network outside the panels, using the predicted SDF. We will clarify the paper.
>
> 5. *Adding the dimensionality of z to Fig 2.*
>
> We will revise Fig. 2 to include the dimensionality of $\textbf{z}$.

---

> > ### Comment · Reviewer_cVr6 · 2023-08-20
> >
> > Thanks to the authors for the clarifications. My concerns are addressed.
> >
> > After reading the thorough discussions with reviewer LvBx, I realized that I wasn't aware of the ULNeF paper. It's true that there are certain similarities of the principles handling multiple layers of clothing -- hence compromising the task-wise novelty by a bit, but the concrete way of doing so is different. For example, the 2D SDF in combination with a 2D-to-3D mapping function in this paper is an interesting formulation. Therefore, I still believe that this paper's technical contributions can inspire future work in this direction and should be accepted.

---

> > > ### Author Response · Authors · 2023-08-20
> > >
> > > Thank you for your feedback and acknowledgement of our contributions.

---

### Official Review · Reviewer_kV1W · 2023-07-07

**Soundness:** 3 good
**Presentation:** 3 good
**Contribution:** 3 good
**Rating:** 6
**Confidence:** 4

**Summary:**

The article introduces a 3D clothing generation and driving method inspired by the traditional clothing production process, which has made certain contributions to the reconstruction of clothing, especially multi-layer clothing. This method can achieve the reconstruction of multiple pieces of clothing, the editing of clothing shape and texture, as well as the driving and collision detection processes of clothing through two-dimensional to three-dimensional mapping. The feasibility of this method can be demonstrated through training and validation on the synthesized dataset.

**Strengths:**

- A method has been proposed to solve the problem of multi-layer clothing threading, converting the three-dimensional threading problem to the two-dimensional template level for processing, improving the efficiency and effectiveness of collision detection, and providing technological innovation for the reconstruction of multi-layer clothing.
- A method flow inspired by the designer's clothing production process has been proposed, providing a new approach for the reconstruction of multi-layer clothing.

**Weaknesses:**

- According to the description in line 226, the training set (400 shirts, 300 skirts, and 200 pants) and the test set (20 shirts, 20 skirts, and 20 pants) of the experiment are both composed of the same three types of clothing. Perhaps fewer types may lead to limitations in the generalization ability of methods?
- According to the description in section 3.1, the style of clothes is controlled by Potential space interpolation. This means that the style type and change direction of clothes seem to be displayed by the size of Potential space, and the existing data size seems difficult to support enough Potential space.
- According to the description in line 96, the stitching method of clothing seems to be pre-defined, which limits the generalization of clothing styles. If more flexible and variable stitching methods can be chosen, it seems that better universality can be achieved.

**Questions:**

According to the description in section 2.5 of the supporting materials, it seems that a potential code is needed to recover the human body from the image. How was this potential code obtained?

**Limitations:**

No extra.

---

> ### Author Rebuttal · Authors · 2023-08-07
>
> We thank you for your acknowledgement of our contribution in cloth modeling and multi-layer clothing. Below are our responses to your comments and questions.
>
> 1. *Fewer types may lead to limitations in the generalization ability.*
>
> We could train on more types if we had the data for them, which would improve the generalization abilities of our model. This expansion is feasible because, in industrial practice, the majority of garments are decomposed into flat sewing patterns, and our representation would effectively apply to such patterns.
>
> 2. *Existing data size seems difficult to support enough Potential space.*
>
> Our dataset is generated using the software of [Korosteleva2021] with sewing pattern parametrization. The specific parameters we use to generate shirts are \{length, width, hem width, collar width, front/back collar depth, sleeve connection width, sleeve opening width, sleeve length\}. We uniformly sample values from pre-defined ranges for these parameters, enabling us to generate the sewing patterns and their corresponding 3D meshes. We follow the same process to generate data for trousers and skirts, each with their own set of parameters. Finally, we get 400 shirts, 300 skirts and 200 pairs of trousers for training, and 20 shirts, 20 skirts and 20 pairs of trousers for testing.
>
> While our current dataset captures variations in style and shape, we acknowledge that it may not cover the entire range of garment shapes and styles due to their vast diversity. However, this can be mitigated by designing and training on additional patterns. As stated above, we ackonwledge that expanding the dataset to encompass a broader range of garment variations would enhance the robustness and generalizability of our method.
>
> Furthermore, as demonstrated in Section 3 of the supplementary material, our approach can be easily extended to sewing patterns with more panels. This implies that our method has the potential to model complex garments beyond those represented in the current dataset. We believe this scalability and adaptability further strengthen the applicability and value of our proposed method.
>
> 3. *The stitching method of clothing seems to be pre-defined.*
>
> Indeed, the stitching information is built-into the sewing patterns, but our implicit formulation allows us to handle garment types with different stitching patterns using a single neural network. The reviewer is right to point out that a more flexible stitching method could potentially enhance the generalization of clothing styles. This is an interesting direction for future research.
>
> 4. *How was the potential code of human body obtained from images?*
>
> We utilize the SMPL model, which incorporates shape parameter $\mathbf{B}$ and pose parameter $\mathbf{\Theta}$, to model the human body in our approach. As stated in Line 268-270 of the main paper, we estimate these parameters $\mathbf{B}$ and $\mathbf{\Theta}$ from the input images using the algorithm proposed by [Rong2021]. The latent code for the garments are randomly initialized and then optimized with gradient descent to fit the observations as Eq. (11).
>
> ### References
>
> *M. Korosteleva and S. Lee. Generating Datasets of 3D Garments with Sewing Patterns. In Advances in Neural Information Processing Systems, 2021.*
> *Y. Rong, T. Shiratori, and H. Joo. Frankmocap: Fast monocular 3d hand and body motion capture by regression and integration. In International Conference on Computer Vision Workshops, 2021.*

---

### Author Rebuttal · Authors · 2023-08-07

We would like to thank all reviewers for their valuable suggestions and constructive comments. We have carefully considered all of the suggestions and concerns raised by each reviewer and responded to each of them below. We will implement these suggestions in our revised paper.

The attached PDF file contains the figure, illustrating the spatial error distribution as suggested by Reviewer 5F53, and the tables presenting the experimental results as recommended by Reviewer LvBx.

Once again, we sincerely thank all reviewers for their expertise and the time they spent in reviewing our paper.

---

### Decision · Program_Chairs · 2023-09-21

**Decision:**

Accept (poster)

**Comment:**

The paper presents an interesting solution to the multi-layer garment draping problem. This is not the first paper exploring this problem setup. The rebuttal helped to clarify several of the concerns raised in the original reviews. In the post-rebuttal discussion, most of the reviewers were confident about the work. There was one negative reviewer, but in the end, the AC championed the paper. The authors engaged in a useful post-rebuttal discussion that helped to highlight the improvements over existing methods.

Please incorporate the additional details (evaluations) in the final version or in the supplementary. The additional experiments are valuable for the readers.